# Epidemic Learning: Boosting Decentralized Learning with Randomized Communication

**Martijn de Vos** *      **Sadegh Farhadkhani** †      **Rachid Guerraoui**
**Anne-Marie Kermarrec**      **Rafael Pires**      **Rishi Sharma**

EPFL, Switzerland

## Abstract

We present Epidemic Learning (EL), a simple yet powerful decentralized learning (DL) algorithm that leverages changing communication topologies to achieve faster model convergence compared to conventional DL approaches. At each round of EL, each node sends its model updates to a *random sample* of $s$ other nodes (in a system of $n$ nodes). We provide an extensive theoretical analysis of EL, demonstrating that its changing topology culminates in superior convergence properties compared to the state-of-the-art (static and dynamic) topologies. Considering smooth non-convex loss functions, the number of transient iterations for EL, *i.e.*, the rounds required to achieve asymptotic linear speedup, is in $\mathcal{O}(n^3/s^2)$ which outperforms the best-known bound $\mathcal{O}(n^3)$ by a factor of $s^2$, indicating the benefit of randomized communication for DL. We empirically evaluate EL in a 96-node network and compare its performance with state-of-the-art DL approaches. Our results illustrate that EL converges up to $1.7\times$ quicker than baseline DL algorithms and attains $2.2\%$ higher accuracy for the same communication volume.

## 1 Introduction

In Decentralized Learning (DL), multiple machines (or nodes) collaboratively train a machine learning model without any central server [32, 37, 42]. Periodically, each node updates the model using its local data, sends its model updates to other nodes, and averages the received model updates, all without sharing raw data. Compared to centralized approaches [28], DL circumvents the need for centralized control, ensures scalability [21, 32], and avoids imposing substantial communication costs on a central server [58]. However, DL comes with its own challenges. The exchange of model updates with all nodes can become prohibitively expensive in terms of communication costs as the network size grows [25]. For this reason, nodes in DL algorithms usually exchange model updates with only a small number of other nodes in a particular round, *i.e.*, they perform partial averaging instead of an All-Reduce (network-wide) averaging of local model updates [57].

A key element of DL algorithms is the communication topology, governing how model updates are exchanged between nodes. The properties of the communication topology are critical for the performance of DL approaches as it directly influences the speed of convergence [27, 54, 56, 52]. The seminal decentralized parallel stochastic gradient descent (D-PSGD) algorithm and many of its proposed variants rely on a static topology, *i.e.*, each node exchanges its model with a set of neighboring nodes that remain fixed throughout the training process [32]. More recent approaches study changing topologies, *i.e.*, topologies that change during training, with notable examples being time-varying graphs [24, 36, 51], one-peer exponential graphs [57], EquiTopo [53], and Gossip Learning [16, 17, 20]. We discuss these works in more detail in Section 5.

---

*Authors are listed in alphabetical order.

†Corresponding author <sadegh.farhadkhani@epfl.ch>.

37th Conference on Neural Information Processing Systems (NeurIPS 2023).

This paper investigates the benefits of *randomized communication* for DL. Randomized communication, extensively studied in distributed computing, has been proven to enhance the performance of fundamental algorithms, including consensus and data dissemination protocols [22, 5, 8]. In the case of DL, randomized communication can reduce the convergence time and therefore communication overhead [11]. In this work, we specifically consider the setting where each node communicates with a random subset of other nodes that changes at each round, as with *epidemic* interaction schemes [12, 38]. We also focus on the scenario where data is unevenly distributed amongst nodes, *i.e.*, non independent and identically distributed (non-IID) settings, a common occurrence in DL [18].

To illustrate the potential of randomized communication, we empirically compare model convergence in Figure 1 for static $s$-regular topologies (referred to as `Static-Topo`) and randomized topologies (referred to as `Rand.-Topo` (EL), our work) in 96-node networks on the CIFAR-10 learning task. As particular static topologies can lead to sub-optimal convergence, we also experiment with the setting in which nodes are stuck in an *unlucky* static $s$-regular topology (referred to as `Static-Topo-Unlucky`). This unlucky topology consists of two network partitions, connected by just two edges. Figure 1 reveals that dynamic topologies converge quicker than static topologies. After 3000 communication rounds, `Rand.-Topo` (EL, our work) achieves 65.3% top-1 test accuracy, compared to 63.4% and 61.8% for Static-Random and Static-Unlucky, respectively. Additional experiments can be found in Section 4, and their setup is elaborated in Appendix C.

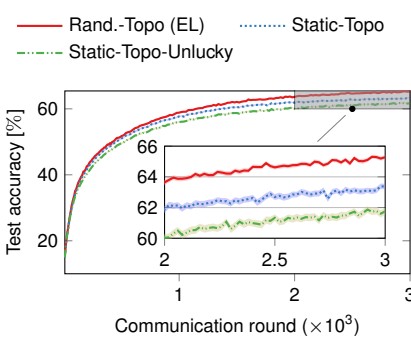

Figure 1: Randomized topologies can converge quicker than static ones.

**Our contributions** Our paper makes the following four contributions:

- We formulate, design, analyze, and experimentally evaluate **Epidemic Learning (EL)**, a novel DL algorithm in which nodes collaboratively train a machine learning model using a dynamically changing, randomized communication topology. More specifically, in EL, at each round, each node sends its model update to a *random sample* of $s$ other nodes (out of $n$ total nodes). This process results in a randomized topology that changes every round.

- We first analyze an EL variant, named **EL-Oracle**, where the union of the random samples by all nodes forms an $s$-regular random graph in each round. EL-Oracle ensures a perfectly balanced communication load among the participating nodes as each node sends and receives exactly $s$ model updates every round. Nevertheless, achieving an $s$-regular graph necessitates coordination among the nodes, which is not ideal in a decentralized setting. To address this challenge, we also analyze another EL variant, named **EL-Local**. In EL-Local, each node independently and locally draws a uniformly random sample of $s$ other nodes at each round and sends its model update to these nodes. We demonstrate that EL-Local enjoys a comparable convergence guarantee as EL-Oracle without requiring any coordination among the nodes and in a fully decentralized manner.

- Our theoretical analysis in Section 3 shows that EL surpasses the best-known static and randomized topologies in terms of convergence speed. More precisely, we prove that EL converges with the rate $\mathcal{O}\left(1/\sqrt{nT} + 1/\sqrt[3]{sT^2} + 1/T\right)$, where $T$ is the number of learning rounds. Similar to most state-of-the-art algorithms for decentralized optimization [30, 24] and centralized stochastic gradient descent (SGD) (*e.g.*, with a parameter server) [29], our rate asymptotically achieves **linear speedup**, *i.e.*, when $T$ is sufficiently large, the first term in the convergence rate $\mathcal{O}(1/\sqrt{nT})$ becomes dominant and improves with respect to the number of nodes.

  Even though linear speedup is a very desirable property, DL algorithms often require many more rounds to reach linear speedup compared to centralized SGD due to the additional error (the second term in the above convergence rate) arising from partial averaging of the local updates. To capture this phenomenon and to compare different decentralized learning algorithms, previous works [10, 50, 53, 57] adopt the concept of **transient iterations** which are the number of rounds before a decentralized algorithm reaches its linear speedup stage, *i.e.*, when $T$ is relatively small such that the second term of the convergence rate dominates the first term.

  We derive that EL requires $\mathcal{O}(n^3/s^2)$ transient iterations, which improves upon the best known bound by a factor of $s^2$. We also show this result in Table 1. We note that while EL matches the state-of-

Table 1: Comparison of EL with state-of-the-art DL approaches (grouped by topology family). We compare EL-Oracle and EL-Local to ring, torus, Erdős–Rényi, exponential and EquiTopo topologies.

| Method | Per-Iter Out Msgs. | Transient Iterations | Topology | Communication |
|---|---|---|---|---|
| **Ring [23]** | 2 | $\mathcal{O}(n^{11})$ | static | undirected |
| **Torus [23]** | 4 | $\mathcal{O}(n^7)$ | static | undirected |
| **E.-R. Rand [45]** | $\mathcal{O}(\log n)$ | $\tilde{\mathcal{O}}(n^3)$ | static | undirected |
| **Static Exp. [57]** | $\log n$ | $\mathcal{O}(n^3 \log^4 n)$ | static | directed |
| **One-Peer Exp. [57]** | 1 | $\mathcal{O}(n^3 \log^4 n)$ | semi-dynamic[1] | directed |
| **D-EquiStatic [53]** | $\log n$ | $\mathcal{O}(n^3)$ | static | directed |
| **U-EquiStatic [53]** | $\log n$ | $\mathcal{O}(n^3)$ | static | undirected |
| **OD-EquiDyn [53]** | 1 | $\mathcal{O}(n^3)$ | semi-dynamic[1] | directed |
| **OU-EquiDyn [53]** | 1 | $\mathcal{O}(n^3)$ | semi-dynamic[1] | undirected |
| **EL-Oracle (ours)** | $s$ | $\mathcal{O}(n^3/s^2)$ | rand.-dynamic[2] | undirected |
| **EL-Local (ours)** | $s$ | $\mathcal{O}(n^3/s^2)$ | rand.-dynamic[2] | directed |

[1] Semi-dynamic topologies remain fixed throughout the learning process but nodes select subsets of adjacent (neighboring) nodes each round to communicate with.
[2] In a randomized-dynamic topology, the topology is replaced each round.

the-art bounds when $s \in \mathcal{O}(1)$, it offers additional flexibility over other methods through parameter $s$ that provably improves the theoretical convergence speed depending on the communication capabilities of the nodes. For instance, when $s \in \mathcal{O}(\log n)$ as in Erdős–Rényi and EquiStatic topologies, the number of transient iterations for EL reduces to $\mathcal{O}(n^3/\log^2 n)$, outperforming other methods. This improvement comes from the fact that the second term in our convergence rate is superior to the corresponding term $\mathcal{O}(1/\sqrt[3]{p^2 T^2})$ in the rate of D-PSGD, where $p \in (0, 1]$ is the spectral gap of the mixing matrix. We expound more on this in Section 3.

- We present in Section 4 our experimental findings. Using two standard image classification datasets, we compare EL-Oracle and EL-Local against static regular graphs and the state-of-the-art EquiTopo topologies. We find that EL-Oracle and EL-Local converge faster than the baselines and save up to $1.7\times$ communication volume to reach the highest accuracy of the most competitive baseline.

## 2 Epidemic Learning

In this section, we first formally define the decentralized optimization problem. Then we outline our EL algorithm and its variants in Section 2.2.

### 2.1 Problem statement

We consider a system of $n$ nodes $[n] := \{1, \ldots, n\}$ where the nodes can communicate by sending messages. Similar to existing work in this domain [53], we consider settings in which a node can communicate with all other nodes. The implications of this assumption are further discussed in Appendix D. Consider a data space $\mathcal{Z}$ and a loss function $f : \mathbb{R}^d \times \mathcal{Z} \to \mathbb{R}$. Given a parameter $x \in \mathbb{R}^d$, a data point $\xi \in \mathcal{Z}$ incurs a loss of value $f(x, \xi)$. Each node $i \in [n]$ has a data distribution $\mathcal{D}^{(i)}$ over $\mathcal{Z}$, which may differ from the data distributions of other nodes. We define the local loss function of $i$ over distribution $\mathcal{D}^{(i)}$ as $f^{(i)}(x) := \mathbb{E}_{\xi \sim \mathcal{D}^{(i)}} [f(x, \xi)]$. The goal is to collaboratively minimize the *global average loss* by solving the following optimization problem:

$$\min_{x \in \mathbb{R}^d} \left[ F(x) := \frac{1}{n} \sum_{i \in [n]} f^{(i)}(x) \right]. \tag{1}$$

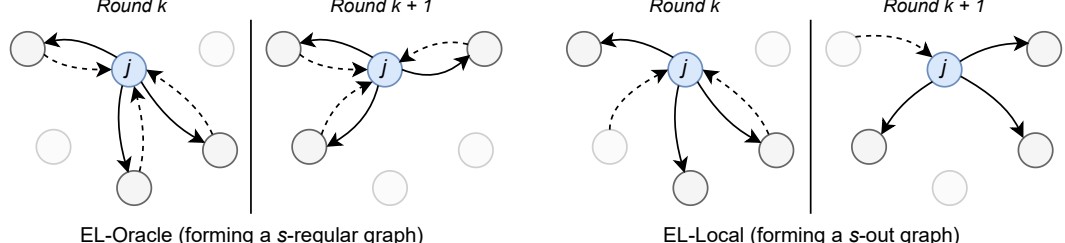

EL-Oracle (forming a s-regular graph)         EL-Local (forming a s-out graph)

Figure 2: EL-Oracle (left) and EL-Local (right), from the perspective of node $j$, with $s = 3$ and for two rounds. We show both outgoing model updates (solid line) and incoming ones (dashed line).

## 2.2 Description of EL

We outline EL, executed by node $i$, in Algorithm 1. We define the initial model of node $i$ as $x_0^{(i)}$ and a step-size $\gamma$ used during the local model update. The EL algorithm runs for $T$ rounds. Each round consists of two phases: a *local update phase* (line 3-5) in which the local model is updated using the local dataset of node $i$, and a *random communication phase* (line 6-9) in which model updates are sent to other nodes chosen randomly. In the local update phase, node $i$ samples a data point $\xi_t^{(i)}$ from its local data distribution $\mathcal{D}^{(i)}$ (line 3), computes the stochastic gradient $g_t^{(i)}$ (line 4) and partially updates its local model $x_{t+1/2}^{(i)}$ using step-size $\gamma$ and gradient $g_t^{(i)}$ (line 5).

The random communication phase follows, where node $i$ first selects $s$ of other nodes from the set of all nodes excluding itself: $[n] \setminus \{i\}$ (line 6). This sampling step is the innovative element of EL, and we present two variants later. It then sends its recently updated local model $x_{t+1/2}^{(i)}$ to the selected nodes and waits for model updates from other nodes. Subsequently, each node $i$ updates its model based on the models it receives from other nodes according to Equation (2). The set of nodes that send their models to node $i$ is denoted by $\mathcal{S}_t^{(i)}$. The new model for node $i$ is computed as a weighted average of the models received from the other nodes and the local model of node $i$, where the weights are inversely proportional to the number of models received plus one.

$$x_{t+1}^{(i)} := \frac{1}{\left|\mathcal{S}_t^{(i)}\right| + 1} \left( x_{t+1/2}^{(i)} + \sum_{j \in \mathcal{S}_t^{(i)}} x_{t+1/2}^{(j)} \right). \tag{2}$$

We now describe two approaches to sample $s$ other nodes (line 6), namely EL-Oracle and EL-Local:

**EL-Oracle** With EL-Oracle, the union of selected communication links forms a $s$-regular topology in which every pair of nodes has an equal probability of being neighbors. Moreover, if node $i$ samples node $j$, $j$ will also sample $i$ (communication is undirected). Figure 2 (left) depicts EL-Oracle sampling from the perspective of node $j$ in two consecutive iterations. One possible way to generate such a dynamic graph is by generating an $s$-regular structure and then distributing a random permutation of

---

**Algorithm 1** Epidemic Learning as executed by a node $i$

---

1: **Require**: Initial model $x_0^{(i)} = x_0 \in \mathbb{R}^d$, number of rounds $T$, step-size $\gamma$, sample size $s$.
2: **for** $t = 0, \ldots, T - 1$ **do**            ▷ Line 3-5: Local training phase
3:      Randomly sample a data point $\xi_t^{(i)}$ from the local data distribution $\mathcal{D}^{(i)}$
4:      Compute the stochastic gradient $g_t^{(i)} := \nabla f(x_t^{(i)}, \xi_t^{(i)})$
5:      Partially update local model $x_{t+1/2}^{(i)} := x_t^{(i)} - \gamma \, g_t^{(i)}$      ▷ Line 6-9: Random communication phase
6:      Sample $s$ other nodes from $[n] \setminus \{i\}$ using EL-Oracle or EL-Local
7:      Send $x_{t+1/2}^{(i)}$ to the selected nodes
8:      Wait for the set of updated models $\mathcal{S}_t^{(i)}$      ▷ $\mathcal{S}_t^{(i)}$ is the set of received models by node $i$ in round $t$
9:      Update $x_{t+1}^{(i)}$ to the average of available updated models according to (2)
10: **end for**

---

nodes at each round. Our implementation (see Section 4) uses a central coordinator to randomize and synchronize the communication topology each round.

**EL-Local** Constructing the $s$-regular topology in EL-Oracle every round can be challenging in a fully decentralized manner as it requires coordination amongst nodes to ensure all nodes have precisely $s$ incoming and outgoing edges. This motivates us to introduce EL-Local, a sampling approach where each node $i$ locally and independently samples $s$ other nodes and sends its model update to them, without these $s$ nodes necessarily sending their model back to $i$. The union of selected nodes now forms a $s$-out topology. Figure 2 (right) depicts EL-Local sampling from the perspective of node $j$ in two consecutive iterations. In practice, EL-Local can be realized either by exchanging peer information before starting the learning process or using a decentralized peer-sampling service that provides nodes with (partial) views on the network [20, 41, 55]. While both the topology construction in EL-Oracle as well as peer sampling in EL-Local add some communication and computation overhead, this overhead is minimal compared to the resources used for model exchange and training.

Even though each node sends $s$ messages for both EL-Oracle and EL-Local, in EL-Local, different nodes may receive different numbers of messages in each training round as each node selects the set of its out-neighbors locally and independent from other nodes. While this might cause an imbalance in the load on individual nodes, we argue that this does not pose a significant issue in practice. To motivate this statement, we run an experiment with $n = 100$, $1000$ and $10\,000$ and for each value of $n$ set $s = \lceil log_2(n) \rceil$. We simulate up to $5000$ rounds for each configuration. We show in Figure 3 a CDF with the number of incoming models each round and observe that the distribution of the number of models received by each node is very light-tailed. In a $10\,000$ node network (and $s = 13$), nodes receive less than 22 models in 99% of all rounds. As such, it is improbable that a node receives

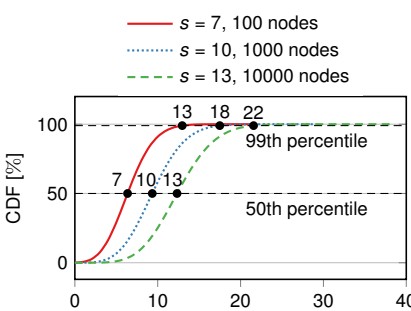

Figure 3: The distribution of incoming models, for different values of $n$ and $s$.

a disproportionally large number of models in a given round. In practice, we can alleviate this imbalance issue by adopting a threshold value $k$ on the number of models processed by each node in a particular round and ignoring incoming models after having received $k$ models already. The sender node can then retry model exchange with another random node that is less occupied.

## 3 Theoretical Analysis

In this section, we first present our main theoretical result demonstrating the finite-time convergence of EL. We then compare our result with the convergence rate of D-PSGD on different topologies.

### 3.1 Convergence of EL

In our analysis, we consider the class of smooth loss functions, and we assume that the variance of the noise of stochastic gradients is bounded. We use of the following assumptions that are classical to the analysis of stochastic first-order methods and hold for many learning problems [4, 14].

**Assumption 1** (Smoothness). *For all $i \in [n]$, the function $f^{(i)} : \mathbb{R}^d \to \mathbb{R}$ is differentiable and there exists $L < \infty$, such that for all $x, y \in \mathbb{R}^d$,*

$$\left\| \nabla f^{(i)}(y) - \nabla f^{(i)}(x) \right\| \le L \left\| y - x \right\|.$$

**Assumption 2** (Bounded stochastic noise). *There exists $\sigma < \infty$ such that for all $i \in [n]$, and $x \in \mathbb{R}^d$,*

$$\mathbb{E}_{\xi \sim \mathcal{D}^{(i)}} \left[ \left\| \nabla f(x, \xi) - f^{(i)}(x) \right\|^2 \right] \le \sigma^2.$$

Moreover, we assume that the heterogeneity among the local loss functions measured by the average distance between the local gradients is bounded.

**Assumption 3** (Bounded heterogeneity). *There exists $\mathcal{H} < \infty$, such that for all $x \in \mathbb{R}^d$,*

$$\frac{1}{n} \sum_{i \in [n]} \left\| \nabla f^{(i)}(x) - \nabla F(x) \right\|^2 \leq \mathcal{H}^2.$$

We note that this assumption is standard in *heterogeneous* (a.k.a. non-i.i.d) settings, *i.e.*, when nodes have different data distributions [32, 57]. In particular, $\mathcal{H}$ can be bounded based on the closeness of the underlying local data distributions [13]. We now present our main theorem.

---

**Theorem 1.** *Consider Algorithm 1. Suppose that assumptions 1, 2 and 3 hold true. Let $\Delta_0$ be a real value such that $F(x_0) - \min_{x \in \mathbb{R}^d} F(x) \leq \Delta_0$. Then, for any $T \geq 1$, $n \geq 2$, and $s \geq 1$:*
*a) For **EL-Oracle**, setting*

$$\gamma \in \Theta\left( \min\left\{ \sqrt{\frac{n\Delta_0}{TL\sigma^2}}, \sqrt[3]{\frac{\Delta_0}{T\alpha_s L^2 (\sigma^2 + \mathcal{H}^2)}}, \frac{1}{L} \right\} \right),$$

*we have*

$$\frac{1}{n} \sum_{i \in [n]} \frac{1}{T} \sum_{t=0}^{T-1} \mathbb{E}\left[ \left\| \nabla F\left( x_t^{(i)} \right) \right\|^2 \right] \in \mathcal{O}\left( \sqrt{\frac{L\Delta_0\sigma^2}{nT}} + \sqrt[3]{\frac{\alpha_s L^2 \Delta_0^2 (\sigma^2 + \mathcal{H}^2)}{T^2}} + \frac{L\Delta_0}{T} \right),$$

*where*

$$\alpha_s := \frac{1}{s+1}\left( 1 - \frac{s}{n-1} \right) \in \mathcal{O}(\tfrac{1}{s}).$$

*b) For **EL-Local**, setting*

$$\gamma \in \Theta\left( \min\left\{ \sqrt{\frac{n\Delta_0}{T(\sigma^2 + \beta_s \mathcal{H}^2)L}}, \sqrt[3]{\frac{\Delta_0}{T\beta_s L^2 (\sigma^2 + \mathcal{H}^2)}}, \frac{1}{L} \right\} \right),$$

*we have*

$$\frac{1}{n} \sum_{i \in [n]} \frac{1}{T} \sum_{t=0}^{T-1} \mathbb{E}\left[ \left\| \nabla F\left( x_t^{(i)} \right) \right\|^2 \right] \in \mathcal{O}\left( \sqrt{\frac{L\Delta_0(\sigma^2 + \beta_s \mathcal{H}^2)}{nT}} + \sqrt[3]{\frac{\beta_s L^2 \Delta_0^2 (\sigma^2 + \mathcal{H}^2)}{T^2}} + \frac{L\Delta_0}{T} \right),$$

*where*

$$\beta_s := \frac{1}{s}\left( 1 - \left(1 - \frac{s}{n-1}\right)^n \right) - \frac{1}{n-1} \in \mathcal{O}(\tfrac{1}{s}).$$

---

To check the tightness of this result, we consider the special case when $s = n-1$. Then, by Theorem 1, we have $\alpha_s = \beta_s = 0$, and thus both of the convergence rates become $\mathcal{O}\left( \sqrt{L\Delta_0\sigma^2/nT} + L\Delta_0/T \right)$, which is the same as the convergence rate of (centralized) SGD for non-convex loss functions [14]. This is expected as, in this case, every node sends its updated model to all other nodes, corresponding to all-to-all communication in a fully-connected topology and thus perfectly averaging the stochastic gradients without any drift between the local models.

The proof of Theorem 1 is given in Appendix A, where we obtain a tighter convergence rate than existing methods. It is important to note that as the mixing matrix of a regular graph is doubly stochastic, for EL-Oracle, one can use the general analysis of D-PSGD with (time-varying) doubly stochastic matrices [24, 30] to obtain a convergence guarantee. However, the obtained rate would not be as tight and would not capture the $\mathcal{O}(1/\sqrt[3]{s})$ improvement in the second term, which is the main advantage of randomization (see Section 3.2). Additionally, it is unclear how these analyses can be generalized to the case where the mixing matrix is not doubly stochastic, which is the case for our EL-Local algorithm. Furthermore, another line of work [1, 43, 44] provides convergence guarantees for decentralized optimization algorithms based on the PushSum algorithm [22], with communicating the mixing weights. It may be possible to leverage this proof technique to prove the convergence of EL-Local. However, this approach yields sub-optimal dimension-dependent convergence guarantees (*e.g.*, see parameter $C$ in Lemma 3 of [1]) and does not capture the benefit of randomized communication.

**Remark 1.** *Most prior work [24, 32] provides the convergence guarantee on the average of the local models $\bar{x}_t = \frac{1}{n}\sum_{i\in[n]} x_t^{(i)}$. However, as nodes cannot access the global averaged model, we provide the convergence rate directly on the local models. Nonetheless, the same convergence guarantee as Theorem 1 also holds for the global averaged model.*

## 3.2 Discussion and comparison to prior results

To provide context for the above result, we note that the convergence rate of decentralized SGD with non-convex loss functions and a doubly stochastic mixing matrix [24, 30] is

$$\mathcal{O}\left(\sqrt{\frac{L\Delta_0\sigma^2}{nT}} + \sqrt[3]{\frac{L^2\Delta_0^2\left(p\sigma^2+\mathcal{H}^2\right)}{p^2T^2}} + \frac{L\Delta_0}{pT}\right), \tag{3}$$

where $p \in (0,1]$ is the spectral gap of the mixing matrix and $1/p$ is bounded by $\mathcal{O}(n^2)$ for ring [23], $\mathcal{O}(n)$ for torus [23], $\mathcal{O}(1)$ for Erdős–Rényi random graph [45], $\mathcal{O}(\log n)$ for exponential graph [57], and $\mathcal{O}(1)$ for EquiTopo [53]. We now compare the convergence of EL against other topologies across two key properties: linear speed-up and transient iterations.

**Linear speed-up** Both of our convergence rates preserve a linear speed-up of $\mathcal{O}(1/\sqrt{nT})$ in the first term. For EL-Oracle, this term is the same as (3). However, in the case of EL-Local, in addition to the stochastic noise $\sigma$, this term also depends on the heterogeneity parameter $\mathcal{H}$ that vanishes when increasing the sample size $s$. This comes from the fact that, unlike EL-Oracle, the communication phase of EL-Local does not preserve the exact average of the local models (*i.e.*, $\sum_{i\in[n]} x_{t+1}^{(i)} \neq \sum_{i\in[n]} x_{t+1/2}^{(i)}$), and it only preserves the average in expectation. This adds an error term to the rate of EL-Local. However, as the update vector remains an unbiased estimate of the average gradient, this additional term does not violate the linear speed-up property. Our analysis suggests that setting $s \approx \frac{\mathcal{H}^2}{\sigma^2}$ can help mitigate the effect of heterogeneity on the convergence of EL-Local. Intuitively, more data heterogeneity leads to more disagreement between the nodes, which requires more communication rounds to converge.

**Transient iterations** Our convergence rates offer superior second and third terms compared to those in (3). This is because first, $p$ can take very small values, particularly when the topology connectivity is low (*e.g.*, $\frac{1}{p} \in \mathcal{O}(n^2)$ for a ring) and second, even when the underlying topology is well-connected and $\frac{1}{p} \in \mathcal{O}(1)$, such as in EquiTopo [53], the second term in our rates still outperforms the one in (3) by a factor of $\sqrt[3]{s}$. This improvement is reflected in the number of transient iterations before the linear speed-up stage, *i.e.*, the number of rounds required for the first term of the convergence rate to dominate the second term [57]. In our rates, the number of transient iterations is in $\mathcal{O}(n^3/s^2)$, whereas in (3), it is $\mathcal{O}(n^3/p^2)$ for the homogeneous case and $\mathcal{O}(n^3/p^4)$ for the heterogeneous case. We remark that $p \in (0,1]$, but $s \geq 1$ is an integer; therefore, when $s \in \mathcal{O}(1)$ the number of transient iterations for EL matches the state-of-the-art bound. However, it can be provably improved by increasing $s$ depending on the communication capabilities of the nodes, which adds more flexibility to EL with theoretical guarantees compared to other methods. For instance, for $s \in \mathcal{O}(\log n)$ as in Erdős–Rényi and EquiStatic topologies, the number of transient iterations for EL becomes $\mathcal{O}(n^3/\log^2 n)$ which outperforms other methods (also see Table 1). Crucially, a key implication of this result is that our algorithm requires fewer rounds and, therefore, less communication to converge. We empirically show the savings in the communication of EL in Section 4.

## 4 Evaluation

We present here the empirical evaluation of EL and compare it with state-of-the-art DL baselines. We first describe the experimental setup and then show the performance of EL-Oracle and EL-Local.

### 4.1 Experimental setup

**Network setup and implementation** We deploy 96 DL nodes for each experiment, interconnected according to the evaluated topologies. When experimenting with *s*-regular topologies, each node maintains a fixed degree of $\lceil log_2(n) \rceil$, *i.e.*, each node has 7 neighbors. For EL-Oracle we introduce a centralized coordinator (oracle) that generates a random 7-Regular topology at the start of each

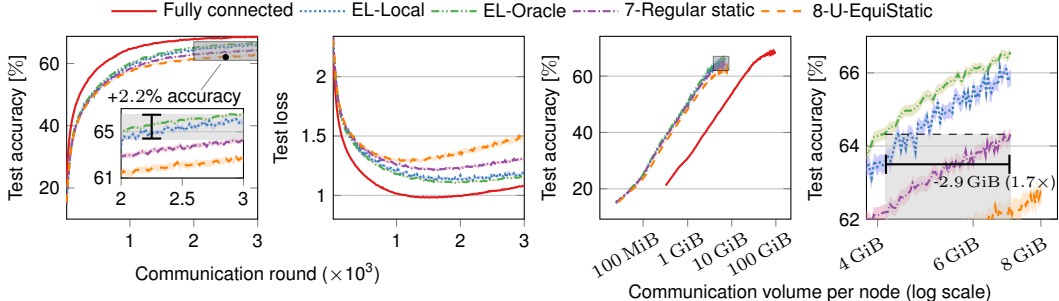

Figure 4: Communication rounds vs. top-1 test accuracy and (left) and communication volume per node vs. test accuracy (right) for the CIFAR-10 dataset.

round and informs all nodes about their neighbors for the upcoming round. For EL-Local we make each node aware of all other nodes at the start of the experiment. To remain consistent with other baselines, we fix $s = \lceil log_2(n) \rceil = 7$ when experimenting with EL, *i.e.*, each node sends model updates to 7 other nodes each round. Both EL-Oracle and EL-Local were implemented using the `DecentralizePy` framework [11] and Python 3.8[3]. For reproducibility, a uniform seed was employed for all pseudo-random generators within each node.

**Baselines** We compare the performance of EL-Oracle and EL-Local against three variants of D-PSGD. Our first baseline is a fully-connected topology (referred to as `Fully connected`), which presents itself as the upper bound for performance given its optimal convergence rate [1]. We also compare with a *s*-regular static topology, the non-changing counterpart of EL-Oracle (referred to as `7-Regular static`). This topology is randomly generated at the start of each run according to the random seed, but is kept fixed during the learning. Finally, we compare EL against the communication-efficient topology U-EquiStatic [53]. Since U-EquiStatic topologies can only have even degrees, we generate U-EquiStatic topologies with a degree of 8 to ensure a fair comparison. We refer to this setting as `8-U-EquiStatic`.

**Learning task and partitioning** We evaluate the baseline algorithms using the CIFAR-10 image classification dataset [26] and the FEMNIST dataset, the latter being part of the LEAF benchmark [7]. In this section we focus on the results for CIFAR-10 and present the results for FEMNIST in Appendix C.4. We employ a non-IID data partitioning using the Dirichlet distribution function [19], parameterized with $\alpha = 0.1$. We use a GN-LENET convolutional neural network [18]. Full details on our experimental setup and hyperparameter tuning can be found in Appendix C.1.

**Metrics** We measure the average top-1 test accuracy and test loss of the model on the test set in the CIFAR-10 learning task every 20 communication rounds. Furthermore, we present the average top-1 test accuracy against the cumulative outgoing communication per node in bytes. We also emphasize the number of communication rounds taken by EL to reach the best top-1 test accuracy of static 7-Regular topology. We run each experiment five times with different random seeds, and we present the average metrics with a 95% confidence interval.

### 4.2 EL against baselines

Figure 4 shows the performance of EL-Oracle and EL-Local against the baselines for the CIFAR-10 dataset. D-PSGD over a fully-connected topology achieves the highest accuracy, as expected, but incurs more than an order of magnitude of additional communication. EL-Oracle converges faster than its static counterparts of `7-Regular static` and `8-U-EquiStatic`. After 3000 communication rounds, EL-Oracle and EL-Local achieve up to 2.2% higher accuracy compared to `7-Regular static` (the most competitive baseline). Moreover, EL-Oracle takes up to $1.7\times$ fewer communication rounds and saves 2.9 GiB of communication to reach the best accuracy attained by 7-Regular static. Surprisingly, `8-U-EquiStatic` shows worse performance compared to `7-Regular static`. The plots further highlight that the less constrained EL-Local variant has a very competitive performance compared to EL-Oracle: there is negligible utility loss when sampling locally compared to generating a *s*-Regular graph every round. We provide additional observations and results in Appendix C.

---

[3]Source code can be found at https://github.com/sacs-epfl/decentralizepy/releases/tag/epidemic-neurips-2023.

Table 2: A summary of key experimental findings for the CIFAR-10 dataset.

| Topology | Top-1 Test Accuracy (%) | Top-1 Test Loss | Communication to Target Accuracy (GiB) |
|---|---|---|---|
| **Fully connected** | 68.67 | 0.98 | 30.54 |
| **7-Regular static** | 64.32 | 1.21 | 7.03 |
| **8-U-EquiStatic**[1] | 62.72 | 1.28 | - |
| **EL-Oracle** | 66.56 | 1.10 | 4.12 |
| **EL-Local** | 66.14 | 1.12 | 4.37 |

[1] The 8-U-EquiStatic topology did not reach the 64.32% target accuracy.

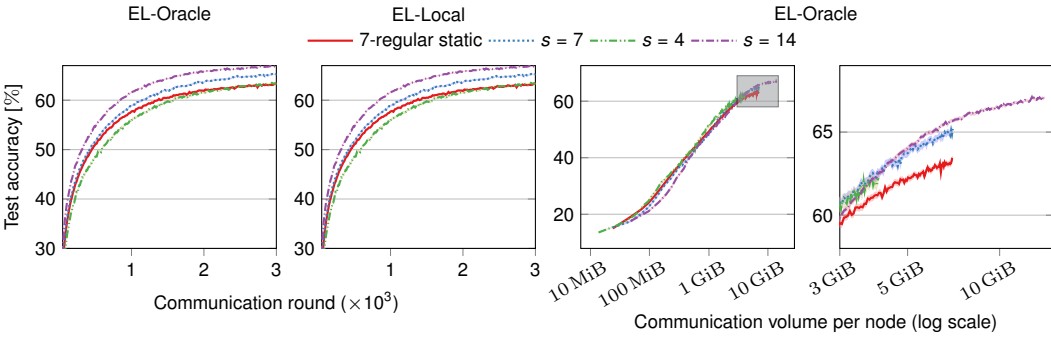

Figure 5: The test accuracy of EL-Oracle and EL-Local (left) and communication volume per node of EL-Oracle (right) for different value of sample size $s$ and a 7-Regular static topology.

We summarize our main experimental findings in Table 2, which outlines the highest achieved top-1 test accuracy, lowest top-1 test loss, and communication cost to a particular target accuracy for our evaluated baselines. This target accuracy is chosen as the best top-1 test accuracy achieved by the 7-Regular static topology (64.32%), and the communication cost to reach this target accuracy is presented for all the topologies in the right-most column of Table 2. In summary, EL-Oracle and EL-Local converge faster and to higher accuracies compared to 7-Regular static and 8-U-EquiStatic topologies, and require $1.7\times$ and $1.6\times$ less communication cost to reach the target accuracy, respectively.

### 4.3 Sensitivity Analysis of sample size $s$

The sample size $s$ determines the number of outgoing neighbours of each node at each round of EL-Oracle and EL-Local. We show the impact of this parameter on the test accuracy in the two left-most plots in Figure 5, for varying values of $s$, against the baseline of a 7-regular static graph. We chose the values of $s$ as $\lceil \frac{log_2(n)}{2} \rceil = 4$, and $\lceil 2log_2(n) \rceil = 14$, over and above $\lceil log_2(n) \rceil = 7$. We observe that, as expected, increasing $s$ leads to quicker convergence for both EL-Oracle and EL-Local (see Theorem 1). Increasing $s$ also directly increases the communication volume. The two right-most plots in Figure 5 show the test accuracy of EL-Oracle when the communication volume increases. After 3000 communication rounds, EL-Oracle with $s = 14$ has incurred a communication volume of 15.1 GiB, compared to 4.3 GiB for $s = 4$. An optimal value of $s$ depends on the network of the environment where EL is deployed. In data center settings where network links usually have high capacities, one can employ a high value of $s$. In edge settings with limited network capacities, however, the value of $s$ should be smaller to avoid network congestion.

## 5 Related Work

**Decentralized Parallel Stochastic Gradient Descent (D-PSGD)** Stochastic Gradient Descent is a stochastic variant of the gradient descent algorithm and is widely used to solve optimization problems at scale [14, 39, 46]. Decentralized algorithms using SGD have gained significant adoption as a method to train machine learning models, with Decentralized Parallel SGD (D-PSGD) being the most well-known DL algorithm [15, 31–33, 49]. D-PSGD avoids a server by relying on a communication topology describing how peers exchange their model updates [32].

**Static topologies** In a static topology, all nodes and edges remain fixed throughout the training process. The convergence speed of decentralized optimization algorithms closely depends on the mixing properties of the underlying topologies. There is a large body of work [2, 3, 6, 40] studying the mixing time of different random graphs such as the Erdos-Renyi graph and the geometric random graph. As the Erdos-Renyi graphs have better mixing properties [45], we compare EL with this family in Table 1. In [57], the authors prove that static exponential graphs in which nodes are connected to $\mathcal{O}(\log n)$ neighbors are an effective topology for DL. EquiStatic is a static topology family whose consensus rate is independent of the network size [53].

**Semi-dynamic topologies** Several DL algorithms impose a fixed communication topology at the start of the training but have nodes communicate with random subsets of their neighbors each round. We classify such topologies as semi-dynamic topologies. The one-peer exponential graph has each node cycling through their $\mathcal{O}(\log n)$ neighbors and has a similar convergence rate to static exponential graphs [57]. In the AD-PSGD algorithm, each node independently trains and averages its model with the model of a randomly selected neighbour [33]. EquiDyn is a dynamic topology family whose consensus rate is independent of the network size [53].

**Time-varying and randomized topologies** A time-varying topology is a topology that changes throughout the training process [9]. The convergence properties of time-varying graphs in distributed optimization have been studied by various works [23, 35, 36, 43, 45]. While these works provide convergence guarantees for decentralized optimization algorithms over time-varying (or random) topologies, they do not show the superiority of randomized communication, and they do not prove a convergence rate that cannot be obtained with a static graph [53]. Another work [34] considers client subsampling in decentralized optimization where at each round, only a fraction of nodes participate in the learning procedure. This approach is orthogonal to the problem we consider in this paper.

**Gossip Learning** Closely related to EL-Local is gossip learning (GL), a DL algorithm in which each node progresses through rounds independently from other peers [48]. In each round, a node sends their model to another random node and aggregates incoming models received by other nodes, weighted by age. While GL shows competitive performance compared to centralized approaches [16, 17], its convergence on non-convex loss functions has not been theoretically proven yet [47]. While at a high level, EL-Local with $s = 1$ may look very similar to GL, there are some subtle differences. First, GL operates in an asynchronous manner whereas EL proceeds in synchronous rounds. Second, GL applies weighted averaging when aggregating models, based on model age, and EL aggregates models unweighted. Third, if a node in GL receives multiple models in a round, this node will aggregate the received model with its local model for each received model separately, whereas in EL, there will be a single model update per round, and all the received models from that round are aggregated together.

In contrast to the existing works, EL leverages a fully dynamic and random topology that changes each round. While static and semi-dynamic topologies have shown to be effective in certain settings, EL surpasses their performance by enabling faster convergence, both in theory and in practice.

## 6 Conclusions

We have introduced Epidemic Learning (EL), a new DL algorithm that accelerates model convergence and saves communication costs by leveraging randomized communication. The key idea of EL is that in each round, each node samples $s$ other nodes in a $n$-node network and sends their model updates to these sampled nodes. We introduced two variants of the sampling approach: EL-Oracle in which the communication topology forms a $s$-regular graph each round, and EL-Local which forms a $s$-out graph. We theoretically proved the convergence of both EL variants and derived that the number of transient iterations for EL is in $\mathcal{O}(n^3/s^2)$, outperforming the best-known bound $\mathcal{O}(n^3)$ by a factor of $s^2$ for smooth non-convex functions. Our experimental evaluation on the CIFAR-10 learning task showcases the $1.7\times$ quicker convergence of EL compared to baselines and highlights the reduction in communication costs.

## Acknowledgement

This work has been supported in part by the Swiss National Science Foundation (SNSF) project 200021-200477.

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

# Appendix

## Organization

The appendices are organized as follows:

- Appendix A proves the convergence guarantee of EL.

- In Appendix B, we prove the key lemmas that are used in the convergence proof.

- Appendix C provides experimental details, computational resources used, and further evaluation of EL with independent and identically distributed (IID) data distributions and the FEMNIST dataset.

- In Appendix D, we discuss the effect of network connectivity on the performance of EL.

## A   Convergence Proof

In this section, we prove Theorem 1, by setting

$$\gamma := \min \left\{ \sqrt{\frac{n\Delta_0}{TL\sigma^2}}, \sqrt[3]{\frac{\Delta_0}{100T\alpha_s L^2 (\sigma^2 + \mathcal{H}^2)}}, \frac{1}{20L} \right\}, \tag{4}$$

for EL-Oracle, and

$$\gamma := \min \left\{ \sqrt{\frac{n\Delta_0}{T (211\sigma^2 + 332\beta_s \mathcal{H}^2) L}}, \sqrt[3]{\frac{\Delta_0}{250T\beta_s L^2 (\sigma^2 + \mathcal{H}^2)}}, \frac{1}{20L} \right\}, \tag{5}$$

for EL-Local.

**Notation:** For any set of $n$ vectors $\{x^{(1)}, \ldots, x^{(n)}\}$, we denote their average by $\bar{x} := \frac{1}{n} \sum_{i \in [n]} x^{(i)}$.

### A.1   Proof steps

We outline here the critical elements for proving Theorem 1.

**Mixing efficiency of EL.**  First, we analyze the mixing properties of the communication phase of both variants of the algorithm, and we observe that:

(a) while EL-Oracle preserves the exact average of the local vectors, i.e., $\bar{x}_{t+1} = \bar{x}_{t+1/2}$, EL-Local only does so in expectation. This property is critical for ensuring that the global update is an unbiased estimator of the average of local gradients, which is necessary for obtaining convergence guarantees with linear speed-up.

(b) The communication phase of both EL-Local and EL-Oracle reduce the drift among the local models by a factor of $\mathcal{O}(1/s)$.

(c) The variance of the averaged model after the communication phase, in EL-Local is in $\mathcal{O}(1/ns)$.

More formally, we have the following lemma.

**Lemma 1.** *Consider Algorithm 1. Let $n \geq 2$, $s \geq 1$, $T \geq 1$, and $t \in \{0, \dots, T-1\}$.*

    *1. For **EL-Oracle**, we have*

        *(a) $\bar{x}_{t+1} = \bar{x}_{t+1/2}$,*

        *(b) $\frac{1}{n^2} \sum_{i,j \in [n]} \mathbb{E}\left[\left\|x_{t+1}^{(i)} - x_{t+1}^{(j)}\right\|^2\right] \leq \alpha_s \cdot \frac{1}{n^2} \sum_{i,j \in [n]} \mathbb{E}\left[\left\|x_{t+1/2}^{(i)} - x_{t+1/2}^{(j)}\right\|^2\right]$,*

    *where*

$$\alpha_s := \frac{1}{s+1}\left(1 - \frac{s}{n-1}\right) \in \mathcal{O}(\frac{1}{s}).$$

    *2. For **EL-Local**, we have:*

        *(a) $\mathbb{E}\left[\bar{x}_{t+1}\right] = \mathbb{E}\left[\bar{x}_{t+1/2}\right]$ (Note that we do not necessarily have $\bar{x}_{t+1} = \bar{x}_{t+1/2}$),*

        *(b) $\frac{1}{n^2} \sum_{i,j \in [n]} \mathbb{E}\left[\left\|x_{t+1}^{(i)} - x_{t+1}^{(j)}\right\|^2\right] \leq \beta_s \cdot \frac{1}{n^2} \sum_{i,j \in [n]} \mathbb{E}\left[\left\|x_{t+1/2}^{(i)} - x_{t+1/2}^{(j)}\right\|^2\right]$,*

        *(c) $\mathbb{E}\left[\left\|\bar{x}_{t+1} - \bar{x}_{t+1/2}\right\|^2\right] \leq \frac{\beta_s}{2n} \cdot \frac{1}{n^2} \sum_{i,j \in [n]} \mathbb{E}\left[\left\|x_{t+1/2}^{(i)} - x_{t+1/2}^{(j)}\right\|^2\right]$,*

    *where*

$$\beta_s := \frac{1}{s}\left(1 - \left(1 - \frac{s}{n-1}\right)^n\right) - \frac{1}{n-1} \in \mathcal{O}(\frac{1}{s})$$

**Uniform bound on model drift and gradient drift.** Next, using the previous lemma, we prove that, for a careful choice of the step-size $\gamma$, the local models stay close to each other during the learning process.

**Lemma 2.** *Suppose that assumptions 1, 2, and 3 hold true. Consider Algorithm 1. Consider a step-size $\gamma$ such that $\gamma \leq \frac{1}{20L}$. For any $t \geq 0$, we obtain that*

$$\frac{1}{n^2} \sum_{i,j \in [n]} \mathbb{E}\left[\left\|x_t^{(i)} - x_t^{(j)}\right\|^2\right] \leq 20\frac{1 + 3\eta_s}{(1 - \eta_s)^2}\eta_s \gamma^2 \left(\sigma^2 + \mathcal{H}^2\right),$$

*and*

$$\frac{1}{n^2} \sum_{i,j \in [n]} \mathbb{E}\left[\left\|g_t^{(i)} - g_t^{(j)}\right\|^2\right] \leq 15 \left(\sigma^2 + \mathcal{H}^2\right),$$

*where $\eta_s = \alpha_s$ for EL-Oracle and $\eta_s = \beta_s$ for EL-Local as defined in Lemma 1.*

**Bound on the gradient norm.** Next, we obtain a bound on the gradient of the loss function $F(x)$ by analyzing the growth of $F(x)$, over the trajectory of the average of local models in Algorithm 1.

**Lemma 3.** *Suppose that assumptions 1 and 2 hold true. Consider Algorithm 1 with $\gamma \leq \frac{1}{2L}$. For any $t \in \{0, \dots, T-1\}$, we obtain that*

$$\mathbb{E}\left[\left\|\nabla F\left(\bar{x}_t\right)\right\|^2\right] \leq \frac{2}{\gamma} \mathbb{E}\left[F(\bar{x}_t) - F(\bar{x}_{t+1})\right] + \frac{L^2}{2n^2} \sum_{i,j \in [n]} \mathbb{E}\left[\left\|x_t^{(i)} - x_t^{(j)}\right\|^2\right]$$

$$+ 2L\gamma\frac{\sigma^2}{n} + \frac{2L}{\gamma}\mathbb{E}\left[\left\|\bar{x}_{t+1} - \bar{x}_{t+1/2}\right\|^2\right].$$

## A.2 Proof of Theorem 1

We can now prove the theorem using the above lemmas.

*Proof of Theorem 1.* By Young's inequality, for any $i \in [n]$, we have

$$\mathbb{E}\left[\left\|\nabla F\left(x_t^{(i)}\right)\right\|^2\right] \leq 2\,\mathbb{E}\left[\|\nabla F(\bar{x}_t)\|^2\right] + 2\,\mathbb{E}\left[\left\|\nabla F(\bar{x}_t) - \nabla F\left(x_t^{(i)}\right)\right\|^2\right]$$

$$\leq 2\,\mathbb{E}\left[\|\nabla F(\bar{x}_t)\|^2\right] + 2L^2\,\mathbb{E}\left[\left\|\bar{x}_t - x_t^{(i)}\right\|^2\right],$$

where in the second inequality we used Assumption 1. Then, averaging over $i \in [n]$, we obtain that

$$\frac{1}{n}\sum_{i\in[n]}\mathbb{E}\left[\left\|\nabla F\left(x_t^{(i)}\right)\right\|^2\right] \leq 2\,\mathbb{E}\left[\|\nabla F(\bar{x}_t)\|^2\right] + 2L^2\frac{1}{n}\sum_{i\in[n]}\mathbb{E}\left[\left\|\bar{x}_t - x_t^{(i)}\right\|^2\right]$$

$$= 2\,\mathbb{E}\left[\|\nabla F(\bar{x}_t)\|^2\right] + L^2\frac{1}{n^2}\sum_{i,j\in[n]}\mathbb{E}\left[\left\|x_t^{(i)} - x_t^{(j)}\right\|^2\right],$$

where we used Lemma 5. Combining this with Lemma 3, we obtain that[4]

$$\frac{1}{n}\sum_{i\in[n]}\mathbb{E}\left[\left\|\nabla F\left(x_t^{(i)}\right)\right\|^2\right] \leq \frac{4}{\gamma}\,\mathbb{E}\left[F(\bar{x}_t) - F(\bar{x}_{t+1})\right] + \frac{2L^2}{n^2}\sum_{i,j\in[n]}\mathbb{E}\left[\left\|x_t^{(i)} - x_t^{(j)}\right\|^2\right]$$

$$+ 4L\gamma\frac{\sigma^2}{n} + \frac{4L}{\gamma}\,\mathbb{E}\left[\left\|\bar{x}_{t+1} - \bar{x}_{t+1/2}\right\|^2\right]. \tag{6}$$

We now analyze the two variants of the algorithm separately.

**EL-Oracle:**
In this case, by Property (2a) of Lemma 1, we have $\bar{x}_{t+1} - \bar{x}_{t+1/2} = 0$, therefore, by (6), we have

$$\frac{1}{n}\sum_{i\in[n]}\mathbb{E}\left[\left\|\nabla F\left(x_t^{(i)}\right)\right\|^2\right] \leq \frac{4}{\gamma}\,\mathbb{E}\left[F(\bar{x}_t) - F(\bar{x}_{t+1})\right] + \frac{2L^2}{n^2}\sum_{i,j\in[n]}\mathbb{E}\left[\left\|x_t^{(i)} - x_t^{(j)}\right\|^2\right] + 4L\gamma\frac{\sigma^2}{n} \tag{7}$$

Note also that by Lemma 1, we have

$$\frac{1}{n^2}\sum_{i,j\in[n]}\mathbb{E}\left[\left\|x_t^{(i)} - x_t^{(j)}\right\|^2\right] \leq 20\frac{1+3\alpha_s}{(1-\alpha_s)^2}\alpha_s\gamma^2\left(\sigma^2 + \mathcal{H}^2\right),$$

Moreover, as shown in Remark 2, we have $\alpha_s \leq 1/2$, and thus, $20\frac{1+3\alpha_s}{(1-\alpha_s)^2} \leq 200$. Therefore,

$$\frac{1}{n^2}\sum_{i,j\in[n]}\mathbb{E}\left[\left\|x_t^{(i)} - x_t^{(j)}\right\|^2\right] \leq 200\alpha_s\gamma^2\left(\sigma^2 + \mathcal{H}^2\right),$$

Combining this with (7), we obtain that

$$\frac{1}{n}\sum_{i\in[n]}\mathbb{E}\left[\left\|\nabla F\left(x_t^{(i)}\right)\right\|^2\right] \leq \frac{4}{\gamma}\,\mathbb{E}\left[F(\bar{x}_t) - F(\bar{x}_{t+1})\right] + 400\alpha_s L^2\gamma^2\left(\sigma^2 + \mathcal{H}^2\right) + 4L\gamma\frac{\sigma^2}{n}$$

Averaging over $t \in \{0, \ldots, T-1\}$, we obtain that

$$\frac{1}{nT}\sum_{i\in[n]}\sum_{t=0}^{T-1}\mathbb{E}\left[\left\|\nabla F\left(x_t^{(i)}\right)\right\|^2\right] \leq \frac{4}{\gamma T}\,\mathbb{E}\left[F(\bar{x}_0) - F(\bar{x}_T)\right] + 400\alpha_s L^2\gamma^2\left(\sigma^2 + \mathcal{H}^2\right) + 4L\gamma\frac{\sigma^2}{n}$$

$$\leq \frac{4}{\gamma T}\Delta_0 + 400\alpha_s L^2\gamma^2\left(\sigma^2 + \mathcal{H}^2\right) + 4L\gamma\frac{\sigma^2}{n}, \tag{8}$$

where we used the fact that $F(\bar{x}_0) - F(\bar{x}_T) \leq F(\bar{x}_0) - \min_{x\in\mathbb{R}^d} F(x) \leq \Delta_0$. Setting

$$\gamma := \min\left\{\sqrt{\frac{n\Delta_0}{TL\sigma^2}}, \sqrt[3]{\frac{\Delta_0}{100T\alpha_s L^2\left(\sigma^2 + \mathcal{H}^2\right)}}, \frac{1}{20L}\right\}, \tag{9}$$

---
[4]In order to obtain a bound on $\mathbb{E}\left[\|\nabla F(\bar{x}_t)\|^2\right]$, we can skip this step and directly use Lemma 3.

as $\frac{1}{\min\{a,b\}} = \max\{\frac{1}{a}, \frac{1}{b}\}$ for any $a, b \geq 0$, we have

$$\frac{1}{\gamma} = \max\left\{\sqrt{\frac{TL\sigma^2}{n\Delta_0}}, \sqrt[3]{\frac{100T\alpha_s L^2 (\sigma^2 + \mathcal{H}^2)}{\Delta_0}}, 20L\right\} \leq \sqrt{\frac{TL\sigma^2}{n\Delta_0}} + \sqrt[3]{\frac{100T\alpha_s L^2 (\sigma^2 + \mathcal{H}^2)}{\Delta_0}} + 20L,$$
(10)

Plugging (9) and (10) in (8), we obtain that

$$\frac{1}{nT}\sum_{i\in[n]}\sum_{t=0}^{T-1}\mathbb{E}\left[\left\|\nabla F\left(x_t^{(i)}\right)\right\|^2\right] \leq 8\sqrt{\frac{L\Delta_0\sigma^2}{nT}} + 38\sqrt[3]{\frac{\alpha_s L^2 \Delta_0^2 (\sigma^2 + \mathcal{H}^2)}{T^2}} + \frac{80L\Delta_0}{T}$$

$$\in \mathcal{O}\left(\sqrt{\frac{L\Delta_0\sigma^2}{nT}} + \sqrt[3]{\frac{\alpha_s L^2 \Delta_0^2 (\sigma^2 + \mathcal{H}^2)}{T^2}} + \frac{L\Delta_0}{T}\right)$$

where we used the fact that $\frac{800}{100^{(2/3)}} \leq 38$.

**EL-Local:**
The proof is similar to EL-Oracle, up to some additional error terms. By Property (2c) of Lemma 1, we obtain that

$$\mathbb{E}\left[\left\|\bar{x}_{t+1} - \bar{x}_{t+1/2}\right\|^2\right] \leq \frac{\beta_s}{2n} \cdot \frac{1}{n^2}\sum_{i,j\in[n]}\mathbb{E}\left[\left\|x_{t+1/2}^{(i)} - x_{t+1/2}^{(j)}\right\|^2\right]$$

$$= \frac{\beta_s}{2n} \cdot \frac{1}{n^2}\sum_{i,j\in[n]}\mathbb{E}\left[\left\|x_t^{(i)} - \gamma g_t^{(i)} - x_t^{(j)} + \gamma g_t^{(j)}\right\|^2\right]$$
(11)

$$\leq \frac{\beta_s}{n} \cdot \frac{1}{n^2}\sum_{i,j\in[n]}\mathbb{E}\left[\left\|x_t^{(i)} - x_t^{(j)}\right\|^2\right] + \frac{\gamma^2\beta_s}{n} \cdot \frac{1}{n^2}\sum_{i,j\in[n]}\mathbb{E}\left[\left\|g_t^{(i)} - g_t^{(j)}\right\|^2\right],$$
(12)

where in the last inequality we use Young's inequality. Combining this with (6), we obtain that

$$\frac{1}{n}\sum_{i\in[n]}\mathbb{E}\left[\left\|\nabla F\left(x_t^{(i)}\right)\right\|^2\right] \leq \frac{4}{\gamma}\mathbb{E}\left[F(\bar{x}_t) - F(\bar{x}_{t+1})\right] + \frac{2L^2}{n^2}\sum_{i,j\in[n]}\mathbb{E}\left[\left\|x_t^{(i)} - x_t^{(j)}\right\|^2\right]$$

$$+ 4L\gamma\frac{\sigma^2}{n} + \frac{4L}{\gamma}\frac{\beta_s}{n} \cdot \frac{1}{n^2}\sum_{i,j\in[n]}\mathbb{E}\left[\left\|x_t^{(i)} - x_t^{(j)}\right\|^2\right] + \frac{4L\gamma\beta_s}{n} \cdot \frac{1}{n^2}\sum_{i,j\in[n]}\mathbb{E}\left[\left\|g_t^{(i)} - g_t^{(j)}\right\|^2\right]$$

$$= \frac{4}{\gamma}\mathbb{E}\left[F(\bar{x}_t) - F(\bar{x}_{t+1})\right] + (2L^2 + \frac{4L\beta_s}{n\gamma})\frac{1}{n^2}\sum_{i,j\in[n]}\mathbb{E}\left[\left\|x_t^{(i)} - x_t^{(j)}\right\|^2\right]$$

$$+ 4L\gamma\frac{\sigma^2}{n} + \frac{4L\gamma\beta_s}{n} \cdot \frac{1}{n^2}\sum_{i,j\in[n]}\mathbb{E}\left[\left\|g_t^{(i)} - g_t^{(j)}\right\|^2\right]$$
(13)

Note that by Remark 2, for any $s \geq 1$, and $n \geq 2$ we have $\beta_s \leq 1 - \frac{1}{e}$, where $e$ is Euler's number. Therefore,

$$20\frac{1 + 3\beta_s}{(1 - \beta_s)^2} \leq 500.$$

Using this, the first bound in Lemma 2 can be simplified to

$$\frac{1}{n^2}\sum_{i,j\in[n]}\mathbb{E}\left[\left\|x_t^{(i)} - x_t^{(j)}\right\|^2\right] \leq 500\beta_s\gamma^2 (\sigma^2 + \mathcal{H}^2)$$
(14)

Combining this with (13), and the second bound of Lemma 2, we have

$$\frac{1}{n}\sum_{i\in[n]}\mathbb{E}\left[\left\|\nabla F\left(x_t^{(i)}\right)\right\|^2\right] \leq \frac{4}{\gamma}\mathbb{E}\left[F(\bar{x}_t) - F(\bar{x}_{t+1})\right] + 4L\gamma\frac{\sigma^2}{n}\left(1 + 15\beta_s + 500\beta_s^2\right)$$

$$+ 4L\gamma\frac{\mathcal{H}^2}{n}\left(15\beta_s + 500\beta_s^2\right) + L^2 1000\beta_s\gamma^2 (\sigma^2 + \mathcal{H}^2).$$

Taking the average over $t \in \{0, \ldots, T-1\}$, we obtain that

$$\frac{1}{nT} \sum_{t=0}^{T-1} \sum_{i \in [n]} \mathbb{E}\left[\left\|\nabla F\left(x_t^{(i)}\right)\right\|^2\right] \leq \frac{4}{T\gamma}\Delta_0 + 4L\gamma\frac{\sigma^2}{n}\left(1 + 15\beta_s + 500\beta_s^2\right)$$

$$+ 4L\gamma\frac{\mathcal{H}^2}{n}\left(15\beta_s + 500\beta_s^2\right) + L^2 1000\beta_s\gamma^2\left(\sigma^2 + \mathcal{H}^2\right).$$

Noting that $\beta_s < 1 - \frac{1}{e}$ (Remark 2), we have

$$\frac{1}{nT} \sum_{t=0}^{T-1} \sum_{i \in [n]} \mathbb{E}\left[\left\|\nabla F\left(x_t^{(i)}\right)\right\|^2\right] \leq \frac{4}{T\gamma}\Delta_0 + 4\frac{L\gamma}{n}\left(211\sigma^2 + 332\beta_s\mathcal{H}^2\right) + 1000\beta_s L^2\gamma^2\left(\sigma^2 + \mathcal{H}^2\right),$$

(15)

Now, setting

$$\gamma = \min\left\{\sqrt{\frac{n\Delta_0}{T\left(211\sigma^2 + 332\beta_s\mathcal{H}^2\right)L}}, \sqrt[3]{\frac{\Delta_0}{250T\beta_s L^2\left(\sigma^2 + \mathcal{H}^2\right)}}, \frac{1}{20L}\right\}, \quad (16)$$

we have

$$\frac{1}{\gamma} = \max\left\{\sqrt{\frac{T\left(211\sigma^2 + 332\beta_s\mathcal{H}^2\right)L}{n\Delta_0}}, \sqrt[3]{\frac{250T\beta_s L^2\left(\sigma^2 + \mathcal{H}^2\right)}{\Delta_0}}, 20L\right\}$$

$$\leq \sqrt{\frac{T\left(211\sigma^2 + 332\beta_s\mathcal{H}^2\right)L}{n\Delta_0}} + \sqrt[3]{\frac{250T\beta_s L^2\left(\sigma^2 + \mathcal{H}^2\right)}{\Delta_0}} + 20L \quad (17)$$

Plugging (17), and (16) in (15) we obtain that

$$\frac{1}{nT} \sum_{t=0}^{T-1} \sum_{i \in [n]} \mathbb{E}\left[\left\|\nabla F\left(x_t^{(i)}\right)\right\|^2\right] \leq 8\sqrt{\frac{L\Delta_0\left(211\sigma^2 + 332\beta_s\mathcal{H}^2\right)}{nT}} + 51\sqrt[3]{\frac{\beta_s L^2\Delta_0^2\left(\sigma^2 + \mathcal{H}^2\right)}{T^2}} + \frac{80L}{T}\Delta_0$$

$$\in \mathcal{O}\left(\sqrt{\frac{L\Delta_0(\sigma^2 + \beta_s\mathcal{H}^2)}{nT}} + \sqrt[3]{\frac{\beta_s L^2\Delta_0^2\left(\sigma^2 + \mathcal{H}^2\right)}{T^2}} + \frac{L\Delta_0}{T}\right).$$

where we used the fact that $\frac{2000}{250^{(2/3)}} \leq 51$. This concludes the proof. $\qquad\square$

## B  Proof of the main lemmas

**Notation:** Let $\mathcal{P}_t$ represent the history from step 0 to $t$. More precisely, we define

$$\mathcal{P}_t := \left\{x_0^{(i)}, \ldots, x_t^{(i)}; \ i = 1, \ldots, n\right\}.$$

Moreover, we use the notation $\mathbb{E}_t\left[\cdot\right] := \mathbb{E}\left[\cdot \mid \mathcal{P}_t\right]$ to denote the conditional expectation given the history $\mathcal{P}_t$ and $\mathbb{E}\left[\cdot\right]$ to denote the total expectation over the randomness of the algorithm; therefore we have, $\mathbb{E}\left[\cdot\right] := \mathbb{E}_0\left[\cdots \mathbb{E}_T\left[\cdot\right]\right]$.

We first prove two simple useful lemmas.

**Lemma 4.** *for any set of $k$ vectors $x^{(1)}, \ldots, x^{(k)}$, we have*

$$\left\|\sum_{i=1}^k x^{(i)}\right\|^2 \leq k\sum_{i=1}^k \left\|x^{(i)}\right\|^2.$$

*Proof.* By the triangle inequality, we have

$$\left\|\sum_{i=1}^k x^{(i)}\right\| \leq \sum_{i=1}^k \left\|x^{(i)}\right\|.$$

The result then follows by noting that using Jensen's inequality, we have

$$\left(\frac{1}{k}\sum_{i=1}^{k}\left\|x^{(i)}\right\|\right)^2 \leq \frac{1}{k}\sum_{i=1}^{k}\left\|x^{(i)}\right\|^2.$$

$\square$

**Lemma 5.** *For any set $\{x^{(i)}\}_{i\in[n]}$ of $n$ vectors, we have*

$$\frac{1}{n}\sum_{i\in[n]}\left\|x^{(i)} - \bar{x}\right\|^2 = \frac{1}{2}\cdot\frac{1}{n^2}\sum_{i,j\in[n]}\left\|x^{(i)} - x^{(j)}\right\|^2.$$

*Proof.*

$$\frac{1}{n^2}\sum_{i,j\in[n]}\left\|x^{(i)} - x^{(j)}\right\|^2 = \frac{1}{n^2}\sum_{i,j\in[n]}\left\|(x^{(i)} - \bar{x}) - (x^{(j)} - \bar{x})\right\|^2$$

$$= \frac{1}{n^2}\sum_{i,j\in[n]}\left[\left\|x^{(i)} - \bar{x}\right\|^2 + \left\|x^{(j)} - \bar{x}\right\|^2 + 2\left\langle x^{(i)} - \bar{x}, x^{(j)} - \bar{x}\right\rangle\right]$$

$$= \frac{2}{n}\sum_{i,j\in[n]}\left\|x^{(i)} - \bar{x}\right\|^2 + \frac{2}{n^2}\sum_{i\in[n]}\left\langle x^{(i)} - \bar{x}, \sum_{j\in[n]}(x^{(j)} - \bar{x})\right\rangle.$$

Noting that $\sum_{j\in[n]}(x^{(j)} - \bar{x}) = 0$, yields the desired result. $\square$

## B.1 Proof of Lemma 1

**Lemma 1.** *Consider Algorithm 1. Let $n \geq 2$, $s \geq 1$, $T \geq 1$, and $t \in \{0,\ldots,T-1\}$.*

1. *For **EL-Oracle**, we have*

   (a) $\bar{x}_{t+1} = \bar{x}_{t+1/2}$,

   (b) $\frac{1}{n^2}\sum_{i,j\in[n]}\mathbb{E}\left[\left\|x_{t+1}^{(i)} - x_{t+1}^{(j)}\right\|^2\right] \leq \alpha_s \cdot \frac{1}{n^2}\sum_{i,j\in[n]}\mathbb{E}\left[\left\|x_{t+1/2}^{(i)} - x_{t+1/2}^{(j)}\right\|^2\right]$,

   *where*

   $$\alpha_s := \frac{1}{s+1}\left(1 - \frac{s}{n-1}\right) \in \mathcal{O}(\frac{1}{s}).$$

2. *For **EL-Local**, we have:*

   (a) $\mathbb{E}\left[\bar{x}_{t+1}\right] = \mathbb{E}\left[\bar{x}_{t+1/2}\right]$ *(Note that we do not necessarily have $\bar{x}_{t+1} = \bar{x}_{t+1/2}$),*

   (b) $\frac{1}{n^2}\sum_{i,j\in[n]}\mathbb{E}\left[\left\|x_{t+1}^{(i)} - x_{t+1}^{(j)}\right\|^2\right] \leq \beta_s \cdot \frac{1}{n^2}\sum_{i,j\in[n]}\mathbb{E}\left[\left\|x_{t+1/2}^{(i)} - x_{t+1/2}^{(j)}\right\|^2\right]$,

   (c) $\mathbb{E}\left[\left\|\bar{x}_{t+1} - \bar{x}_{t+1/2}\right\|^2\right] \leq \frac{\beta_s}{2n}\cdot\frac{1}{n^2}\sum_{i,j\in[n]}\mathbb{E}\left[\left\|x_{t+1/2}^{(i)} - x_{t+1/2}^{(j)}\right\|^2\right]$,

   *where*

   $$\beta_s := \frac{1}{s}\left(1 - \left(1 - \frac{s}{n-1}\right)^n\right) - \frac{1}{n-1} \in \mathcal{O}(\frac{1}{s})$$

*Proof.* For simplicity of the notation, we make the dependence on $t$ implicit, and we denote by $x^{(i)} := x_{t+1/2}^{(i)}$, the input vector and $y^{(i)} := x_{t+1}^{(i)}$ the output vector of node $i$ for the communication phase, and $\mathcal{S}^{(i)} := \mathcal{S}_t^{(i)}$. Therefore, we have

$$y^{(i)} = \frac{1}{|\mathcal{S}^{(i)}| + 1}\left(x^{(i)} + \sum_{j\in\mathcal{S}^{(i)}} x^{(j)}\right).$$

Moreover, in the proof of this lemma, all the expectations $\mathbb{E}\left[\cdot\right]$ are only with respect to the randomness of the communication phase, i.e., we compute the expectations assuming $x^{(i)}$'s are given for all $i \in [n]$, and $y^{(i)}$'s are computed by a random communication phase. Taking the total expectation of the final expression provided for each property then proves it as stated in the lemma.

First, we consider **EL-Oracle**:

Recall that in EL-Oracle, the communication topology forms a random undirected $s$-regular graph, and for each node $i$, all other nodes have the same probability of being $i$'s neighbor.

**Property (a):**
We have

$$\bar{y} = \frac{1}{n} \sum_{i \in [n]} y^{(i)}$$

$$= \frac{1}{n} \sum_{i \in [n]} \frac{1}{\left|\mathcal{S}^{(i)}\right| + 1} \left( x^{(i)} + \sum_{j \in \mathcal{S}^{(i)}} x^{(j)} \right)$$

$$= \frac{1}{n(s+1)} \sum_{i \in [n]} \left( x^{(i)} + \sum_{j \in \mathcal{S}^{(i)}} x^{(j)} \right)$$

$$= \frac{1}{n(s+1)} \left( \sum_{i \in [n]} x^{(i)} + \sum_{i \in [n]} \sum_{j \in \mathcal{S}^{(i)}} x^{(j)} \right),$$

where we used the fact that in EL-Oracle, each node receives exactly $s$ models. Now as each node also sends its model to $s$ other nodes, we have

$$\bar{y} = \frac{1}{n(s+1)} \left( \sum_{i \in [n]} x^{(i)} + \sum_{i \in [n]} s x^{(i)} \right) = \frac{1}{n} \sum_{i \in [n]} x^{(i)} = \bar{x}.$$

**Property (b):**
We have

$$\frac{1}{n} \sum_{i \in [n]} \mathbb{E}\left[ \left\| y^{(i)} - \bar{x} \right\|^2 \right] = \frac{1}{n} \sum_{i \in [n]} \mathbb{E}\left[ \left\| \frac{1}{\left|\mathcal{S}^{(i)}\right| + 1} \left( x^{(i)} + \sum_{j \in \mathcal{S}^{(i)}} x^{(j)} \right) - \bar{x} \right\|^2 \right]$$

$$= \frac{1}{n(s+1)^2} \sum_{i \in [n]} \mathbb{E}\left[ \left\| (x^{(i)} - \bar{x}) + \sum_{j \in \mathcal{S}^{(i)}} (x^{(j)} - \bar{x}) \right\|^2 \right]$$

$$= \frac{1}{n(s+1)^2} \sum_{i \in [n]} \left\| x^{(i)} - \bar{x} \right\|^2 + \frac{1}{n(s+1)^2} \sum_{i \in [n]} \mathbb{E}\left[ \left\| \sum_{j \in \mathcal{S}^{(i)}} (x^{(j)} - \bar{x}) \right\|^2 \right]$$

$$+ \frac{2}{n(s+1)^2} \sum_{i \in [n]} \mathbb{E}\left[ \left\langle (x^{(i)} - \bar{x}), \sum_{j \in \mathcal{S}^{(i)}} (x^{(j)} - \bar{x}) \right\rangle \right]. \tag{18}$$

Let us define $\mathcal{I}_j^{(i)}$ the indicator function showing whether node $j$ and node $i$ are neighbors, i.e., $\mathcal{I}_j^{(i)} = 1$ if $i$ and $j$ are neighbors and $\mathcal{I}_j^{(i)} = 0$ otherwise. We then have

$$\sum_{i\in[n]} \mathbb{E}\left[\left\|\sum_{j\in\mathcal{S}^{(i)}}(x^{(j)}-\bar{x})\right\|^2\right] = \sum_{i\in[n]} \mathbb{E}\left[\left\|\sum_{j\in[n]\backslash\{i\}}\mathcal{I}_j^{(i)}(x^{(j)}-\bar{x})\right\|^2\right]$$

$$= \sum_{i\in[n]} \mathbb{E}\left[\sum_{j\in[n]\backslash\{i\}}\mathcal{I}_j^{(i)}\left\|x^{(j)}-\bar{x}\right\|^2\right] + \sum_{i\in[n]} \mathbb{E}\left[\sum_{j\neq i}\sum_{k\neq i,k\neq j}\mathcal{I}_j^{(i)}\mathcal{I}_k^{(i)}\left\langle x^{(j)}-\bar{x},\, x^{(k)}-\bar{x}\right\rangle\right]$$

$$= \sum_{i\in[n]}\sum_{j\in[n]\backslash\{i\}} \mathbb{E}\left[\mathcal{I}_j^{(i)}\right]\left\|x^{(j)}-\bar{x}\right\|^2 + \sum_{i\in[n]}\sum_{j\neq i}\sum_{k\neq i,k\neq j} \mathbb{E}\left[\mathcal{I}_j^{(i)}\mathcal{I}_k^{(i)}\right]\left\langle x^{(j)}-\bar{x},\, x^{(k)}-\bar{x}\right\rangle.$$

Now note that by symmetry (all nodes have the same probability of being a neighbor of node $i$).

$$\mathbb{E}\left[\mathcal{I}_j^{(i)}\right] = \frac{s}{n-1}.$$

Similarly $\mathcal{I}_j^{(i)}\mathcal{I}_k^{(i)}$ is only 1 when both $j$ and $k$ are neighbors of $i$, thus

$$\mathbb{E}\left[\mathcal{I}_j^{(i)}\mathcal{I}_k^{(i)}\right] = \frac{s(s-1)}{(n-1)(n-2)}.$$

Therefore,

$$\sum_{i\in[n]} \mathbb{E}\left[\left\|\sum_{j\in\mathcal{S}^{(i)}}(x^{(j)}-\bar{x})\right\|^2\right] = s\sum_{i\in[n]}\left\|x^{(j)}-\bar{x}\right\|^2 + \sum_{i\in[n]}\sum_{j\neq i}\sum_{k\neq i,k\neq j}\frac{s(s-1)}{(n-1)(n-2)}\left\langle x^{(j)}-\bar{x},\, x^{(k)}-\bar{x}\right\rangle$$

$$= s\sum_{i\in[n]}\left\|x^{(j)}-\bar{x}\right\|^2 + \sum_{i\in[n]}\sum_{j\neq i}\frac{s(s-1)}{n-1}\left\langle x^{(i)}-\bar{x},\, x^{(j)}-\bar{x}\right\rangle. \tag{19}$$

Also,

$$\sum_{i\in[n]} \mathbb{E}\left[\left\langle(x^{(i)}-\bar{x}),\, \sum_{j\in\mathcal{S}^{(i)}}(x^{(j)}-\bar{x})\right\rangle\right] = \sum_{i\in[n]}\left\langle(x^{(i)}-\bar{x}),\, \sum_{j\neq i}\mathbb{E}\left[\mathcal{I}_j^{(i)}\right](x^{(j)}-\bar{x})\right\rangle$$

$$= \frac{s}{n-1}\sum_{i\in[n]}\sum_{j\neq i}\left\langle x^{(i)}-\bar{x},\, x^{(j)}-\bar{x}\right\rangle \tag{20}$$

Combining (18), (19), and (20) we obtain that

$$\frac{1}{n}\sum_{i\in[n]} \mathbb{E}\left[\left\|y^{(i)}-\bar{x}\right\|^2\right] = \frac{1+s}{n(s+1)^2}\sum_{i\in[n]}\left\|x^{(i)}-\bar{x}\right\|^2 \tag{21}$$

$$+ \frac{1}{n(s+1)^2}\left(\frac{2s}{n-1}+\frac{s(s-1)}{n-1}\right)\sum_{i\in[n]}\sum_{j\neq i}\left\langle x^{(i)}-\bar{x},\, x^{(j)}-\bar{x}\right\rangle.$$

Also note that

$$\sum_{i\in[n]}\sum_{j\neq i}\left\langle x^{(i)}-\bar{x},\, x^{(j)}-\bar{x}\right\rangle = \sum_{i\in[n]}\left\langle x^{(i)}-\bar{x},\, \sum_{j\neq i}x^{(j)}-\bar{x}\right\rangle = -\sum_{i\in[n]}\left\|x^{(i)}-\bar{x}\right\|^2. \tag{22}$$

Combining (21) and (22), we obtain that

$$\frac{1}{n}\sum_{i\in[n]} \mathbb{E}\left[\left\|y^{(i)}-\bar{x}\right\|^2\right] = \frac{1}{s+1}\left(1-\frac{s}{n-1}\right)\frac{1}{n}\sum_{i\in[n]}\left\|x^{(i)}-\bar{x}\right\|^2 \tag{23}$$

Now note that as $\bar{y}$ is the minimizer of $g(z) := \frac{1}{n} \sum_{i \in [n]} \mathbb{E}\left[\left\|y^{(i)} - z\right\|^2\right]$, we have

$$\frac{1}{n} \sum_{i \in [n]} \mathbb{E}\left[\left\|y^{(i)} - \bar{y}\right\|^2\right] \leq \frac{1}{n} \sum_{i \in [n]} \mathbb{E}\left[\left\|y^{(i)} - \bar{x}\right\|^2\right]. \tag{24}$$

Combining this with (23), and using Lemma 5 yields

$$\frac{1}{n^2} \sum_{i,j \in [n]} \mathbb{E}\left[\left\|y^{(i)} - y^{(j)}\right\|^2\right] \leq \frac{1}{s+1}\left(1 - \frac{s}{n-1}\right) \frac{1}{n^2} \sum_{i,j \in [n]} \mathbb{E}\left[\left\|x^{(i)} - x^{(j)}\right\|^2\right],$$

which is the desired result.

Now, we consider **EL-Local**:

Recall that in EL-Local, each node $i$, locally samples uniformly at random $s$ other nodes and sends its model to them. Denote by $A^{(i)} := \left|\mathcal{S}^{(i)}\right|$ the number of nodes that send their vectors to node $i$, and $\mathcal{I}_j^{(i)}$ the indicator showing whether node $j$ sends its vector to node $i$ or not ($\mathcal{I}_j^{(i)} = 1$ if $j$ sends its model to $i$, and $\mathcal{I}_j^{(i)} = 0$ otherwise). We then have

$$A^{(i)} = \sum_{j \in [n] \setminus \{i\}} \mathcal{I}_j^{(i)},$$

and

$$\mathbb{E}\left[A^{(i)}\right] = s.$$

**Property (a):**
For any node $i$, we have

$$\mathbb{E}\left[y^{(i)}\right] = \mathbb{E}\left[\frac{1}{A^{(i)} + 1}\left(x^{(i)} + \sum_{j \in [n] \setminus \{i\}} \mathcal{I}_j^{(i)} x^{(j)}\right)\right]$$

$$= \mathbb{E}\left[\mathbb{E}\left[\frac{1}{A^{(i)} + 1}\left(x^{(i)} + \sum_{j \in [n] \setminus \{i\}} \mathcal{I}_j^{(i)} x^{(j)}\right) | A^{(i)}\right]\right]$$

$$= \mathbb{E}\left[\frac{1}{A^{(i)} + 1}\left(x^{(i)} + \sum_{j \in [n] \setminus \{i\}} \mathbb{E}\left[\mathcal{I}_j^{(i)} | A^{(i)}\right] x^{(j)}\right)\right]$$

$$\stackrel{(a)}{=} \mathbb{E}\left[\frac{1}{A^{(i)} + 1}\left(x^{(i)} + \frac{A^{(i)}}{n-1} \sum_{j \in [n] \setminus \{i\}} x^{(j)}\right)\right]$$

$$= \mathbb{E}\left[\frac{1}{A^{(i)} + 1}\left(x^{(i)} + \frac{A^{(i)}}{n-1}(n\bar{x} - x^{(i)})\right)\right],$$

where (a) uses the fact that $\mathbb{E}\left[\mathcal{I}_j^{(i)} | A^{(i)}\right] = \frac{A^{(i)}}{n-1}$ as all other $n-1$ nodes have the same probability of sending their model to $i$. Denoting $p = \mathbb{E}\left[\frac{A^{(i)}}{A^{(i)} + 1}\right]$, we obtain that

$$\mathbb{E}\left[y^{(i)}\right] = \frac{pn}{n-1}\bar{x} + \left(1 - \frac{pn}{n-1}\right) x^{(i)}.$$

Averaging over all $i \in [n]$ yields the result.

**Property (b):**
We have

$$\frac{1}{n}\sum_{i\in[n]}\mathbb{E}\left[\left\|y^{(i)}-\bar{x}\right\|^2\right]=\frac{1}{n}\sum_{i\in[n]}\mathbb{E}\left[\left\|\frac{1}{A^{(i)}+1}\left(x^{(i)}+\sum_{j\in[n]\setminus\{i\}}\mathcal{I}_j^{(i)}x^{(j)}\right)-\bar{x}\right\|^2\right]$$

$$=\frac{1}{n}\sum_{i\in[n]}\mathbb{E}\left[\mathbb{E}\left[\left\|\frac{1}{A^{(i)}+1}\left(x^{(i)}+\sum_{j\in[n]\setminus\{i\}}\mathcal{I}_j^{(i)}x^{(j)}\right)-\bar{x}\right\|^2\Big|A^{(i)}\right]\right]$$

$$=\frac{1}{n}\sum_{i\in[n]}\mathbb{E}\left[\mathbb{E}\left[\left\|\frac{1}{A^{(i)}+1}\left((x^{(i)}-\bar{x})+\sum_{j\in[n]\setminus\{i\}}\mathcal{I}_j^{(i)}(x^{(j)}-\bar{x})\right)\right\|^2\Big|A^{(i)}\right]\right]$$

$$=\frac{1}{n}\sum_{i\in[n]}\mathbb{E}\left[\frac{1}{(A^{(i)}+1)^2}\mathbb{E}\left[\left\|x^{(i)}-\bar{x}\right\|^2+\sum_{j\neq i}\mathcal{I}_j^{(i)}\left\|x^{(j)}-\bar{x}\right\|^2\Big|A^{(i)}\right]\right]$$

$$+\frac{1}{n}\sum_{i\in[n]}\mathbb{E}\left[\frac{1}{(A^{(i)}+1)^2}\mathbb{E}\left[2\sum_{j\neq i}\mathcal{I}_j^{(i)}\left\langle x^{(i)}-\bar{x},\,x^{(j)}-\bar{x}\right\rangle+\sum_{j\neq i}\sum_{k\neq i,k\neq j}\mathcal{I}_j^{(i)}\mathcal{I}_k^{(i)}\left\langle x^{(j)}-\bar{x},\,x^{(k)}-\bar{x}\right\rangle\Big|A^{(i)}\right]\right]$$

Taking the expectation inside, we obtain that

$$\frac{1}{n}\sum_{i\in[n]}\mathbb{E}\left[\left\|y^{(i)}-\bar{x}\right\|^2\right]=\frac{1}{n}\sum_{i\in[n]}\mathbb{E}\left[\frac{1}{(A^{(i)}+1)^2}\left(\left\|x^{(i)}-\bar{x}\right\|^2+\sum_{j\neq i}\mathbb{E}\left[\mathcal{I}_j^{(i)}|A^{(i)}\right]\left\|x^{(j)}-\bar{x}\right\|^2\right)\right]$$

$$+\frac{1}{n}\sum_{i\in[n]}\mathbb{E}\left[\frac{1}{(A^{(i)}+1)^2}\left(2\sum_{j\neq i}\mathbb{E}\left[\mathcal{I}_j^{(i)}|A^{(i)}\right]\left\langle x^{(i)}-\bar{x},\,x^{(j)}-\bar{x}\right\rangle\right)\right]$$

$$+\frac{1}{n}\sum_{i\in[n]}\mathbb{E}\left[\frac{1}{(A^{(i)}+1)^2}\left(\sum_{j\neq i}\sum_{k\neq i,k\neq j}\mathbb{E}\left[\mathcal{I}_j^{(i)}\mathcal{I}_k^{(i)}|A^{(i)}\right]\left\langle x^{(j)}-\bar{x},\,x^{(k)}-\bar{x}\right\rangle\right)\right].$$

Now note that $\mathbb{E}\left[\mathcal{I}_j^{(i)}|A^{(i)}\right]$ is the probability that $j$ selects $i$, given that in total $A^{(i)}$ nodes select $i$, thus

$$\mathbb{E}\left[\mathcal{I}_j^{(i)}|A^{(i)}\right]=\frac{A^{(i)}}{n-1}.$$

Similarly $\mathcal{I}_j^{(i)}\mathcal{I}_k^{(i)}$ is only 1 when both $j$ and $k$ select $i$, thus

$$\mathbb{E}\left[\mathcal{I}_j^{(i)}\mathcal{I}_k^{(i)}|A^{(i)}\right]=\frac{A^{(i)}(A^{(i)}-1)}{(n-1)(n-2)}$$

Also, note that

$$\sum_{j\neq i}\left\langle x^{(i)}-\bar{x},\,x^{(j)}-\bar{x}\right\rangle=\left\langle x^{(i)}-\bar{x},\,\sum_{j\neq i}(x^{(j)}-\bar{x})\right\rangle=-\left\|x^{(i)}-\bar{x}\right\|^2,$$

and

$$\sum_{j\neq i}\sum_{k\neq i,k\neq j}\left\langle x^{(j)}-\bar{x},\,x^{(k)}-\bar{x}\right\rangle=\sum_{j\neq i}\left\langle x^{(j)}-\bar{x},\,\sum_{k\neq i,k\neq j}(x^{(k)}-\bar{x})\right\rangle$$

$$=-\sum_{j\neq i}\left\langle x^{(j)}-\bar{x},\,(x^{(i)}-\bar{x})+(x^{(j)}-\bar{x})\right\rangle$$

$$=\left\|x^{(i)}-\bar{x}\right\|^2-\sum_{j\neq i}\left\|x^{(j)}-\bar{x}\right\|^2.$$

Combining all yields

$$
\frac{1}{n}\sum_{i\in[n]}\mathbb{E}\left[\left\|y^{(i)}-\bar{x}\right\|^2\right]=\frac{1}{n}\sum_{i\in[n]}\mathbb{E}\left[\frac{1}{(A^{(i)}+1)^2}\left(\left\|x^{(i)}-\bar{x}\right\|^2+\frac{A^{(i)}}{n-1}\sum_{j\neq i}\left\|x^{(j)}-\bar{x}\right\|^2\right)\right]
$$

$$
+\frac{1}{n}\sum_{i\in[n]}\mathbb{E}\left[\frac{1}{(A^{(i)}+1)^2}\left(-\frac{2A^{(i)}}{n-1}\left\|x^{(i)}-\bar{x}\right\|^2\right)\right]
$$

$$
+\frac{1}{n}\sum_{i\in[n]}\mathbb{E}\left[\frac{1}{(A^{(i)}+1)^2}\cdot\frac{A^{(i)}(A^{(i)}-1)}{(n-1)(n-2)}\left(\left\|x^{(i)}-\bar{x}\right\|^2-\sum_{j\neq i}\left\|x^{(j)}-\bar{x}\right\|^2\right)\right]
$$

$$
=\frac{1}{n}\sum_{i\in[n]}\left\|x^{(i)}-\bar{x}\right\|^2\mathbb{E}\left[\frac{1}{(A^{(i)}+1)^2}\left(1-\frac{2A^{(i)}}{n-1}+\frac{A^{(i)}(A^{(i)}-1)}{(n-1)(n-2)}\right)\right]
$$

$$
+\frac{1}{n}\sum_{i\in[n]}\left(\mathbb{E}\left[\frac{1}{(A^{(i)}+1)^2}\left(\frac{A^{(i)}}{n-1}-\frac{A^{(i)}(A^{(i)}-1)}{(n-1)(n-2)}\right)\right]\sum_{j\neq i}\left\|x^{(j)}-\bar{x}\right\|^2\right)
$$

Now note that by symmetry, the distribution of $A^{(i)}$ is the same as $A^{(j)}$ for any $i,j\in[n]$. Therefore,

$$
\frac{1}{n}\sum_{i\in[n]}\mathbb{E}\left[\left\|y^{(i)}-\bar{x}\right\|^2\right]
$$

$$
=\frac{1}{n}\sum_{i\in[n]}\left\|x^{(i)}-\bar{x}\right\|^2\mathbb{E}\left[\frac{1}{(A^{(1)}+1)^2}\left(1-\frac{2A^{(1)}}{n-1}+\frac{A^{(1)}(A^{(1)}-1)}{(n-1)(n-2)}+A^{(1)}-\frac{A^{(1)}(A^{(1)}-1)}{n-2}\right)\right]
$$

Now note that

$$
1-\frac{2A^{(1)}}{n-1}+\frac{A^{(1)}(A^{(1)}-1)}{(n-1)(n-2)}+A^{(1)}-\frac{A^{(1)}(A^{(1)}-1)}{n-2}=1+A^{(1)}-\frac{{A^{(1)}}^2+A^{(1)}}{n-1}=(1+A^{(1)})(1-\frac{A^{(1)}}{n-1}),
$$

thus

$$
\frac{1}{n}\sum_{i\in[n]}\mathbb{E}\left[\left\|y^{(i)}-\bar{x}\right\|^2\right]=\frac{1}{n}\sum_{i\in[n]}\left\|x^{(i)}-\bar{x}\right\|^2\left(\mathbb{E}\left[\frac{1}{A^{(1)}+1}\right]-\frac{1}{n-1}\cdot\mathbb{E}\left[\frac{A^{(1)}}{A^{(1)}+1}\right]\right)
$$

Now note that as each node $j\neq 1$, independently and uniformly selects a set of $s$ nodes, $A^{(1)}$ has a binomial distribution with parameters $n-1$, and $\frac{s}{n-1}$. Therefore, for $s>0$, we have

$$
\mathbb{E}\left[\frac{1}{A^{(1)}+1}\right]=\sum_{k=0}^{n-1}\frac{1}{k+1}\binom{n-1}{k}\left(\frac{s}{n-1}\right)^k\left(1-\frac{s}{n-1}\right)^{n-1-k}
$$

$$
=\frac{n-1}{sn}\sum_{k=0}^{n-1}\binom{n}{k+1}\left(\frac{s}{n-1}\right)^{k+1}\left(1-\frac{s}{n-1}\right)^{n-1-k}
$$

$$
=\frac{n-1}{sn}\left(1-\left(1-\frac{s}{n-1}\right)^n\right).
$$

Also noting that

$$
\mathbb{E}\left[\frac{A^{(1)}}{A^{(1)}+1}\right]=1-\mathbb{E}\left[\frac{1}{A^{(1)}+1}\right],
$$

we obtain that

$$
\frac{1}{n}\sum_{i\in[n]}\mathbb{E}\left[\left\|y^{(i)}-\bar{x}\right\|^2\right]=\left[\frac{1}{s}\left(1-\left(1-\frac{s}{n-1}\right)^n\right)-\frac{1}{n-1}\right]\frac{1}{n}\sum_{i\in[n]}\left\|x^{(i)}-\bar{x}\right\|^2. \quad (25)
$$

Noting that $\frac{1}{n} \sum_{i \in [n]} \mathbb{E}\left[\left\|y^{(i)} - \bar{y}\right\|^2\right] \leq \frac{1}{n} \sum_{i \in [n]} \mathbb{E}\left[\left\|y^{(i)} - \bar{x}\right\|^2\right]$, from (24), and using Lemma 5, we obtain that

$$
\frac{1}{n^2} \sum_{i,j \in [n]} \mathbb{E}\left[\left\|y^{(i)} - y^{(j)}\right\|^2\right] \leq \left[\frac{1}{s}\left(1 - \left(1 - \frac{s}{n-1}\right)^n\right) - \frac{1}{n-1}\right] \frac{1}{n^2} \sum_{i,j \in [n]} \mathbb{E}\left[\left\|x^{(i)} - x^{(j)}\right\|^2\right],
$$

which is the desired result.

**Property (c):**
Note that

$$
\mathbb{E}\left[\|\bar{y} - \bar{x}\|^2\right] = \mathbb{E}\left[\left\|\frac{1}{n} \sum_{i \in [n]} y^{(i)} - \bar{x}\right\|^2\right]
$$

$$
= \frac{1}{n^2} \sum_{i \in [n]} \mathbb{E}\left[\left\|y^{(i)} - \bar{x}\right\|^2\right] + \frac{1}{n^2} \sum_{i \neq j} \mathbb{E}\left[\left\langle y^{(i)} - \bar{x}, \, y^{(j)} - \bar{x}\right\rangle\right]. \quad (26)
$$

Now recall that

$$
y^{(i)} - \bar{x} = \frac{1}{A^{(i)} + 1}\left((x^{(i)} - \bar{x}) + \sum_{k \in [n]\backslash\{i\}} \mathcal{I}_k^{(i)}(x^{(k)} - \bar{x})\right).
$$

This implies that

$$
\frac{1}{n^2} \sum_{i \neq j} \mathbb{E}\left[\left\langle y^{(i)} - \bar{x}, \, y^{(j)} - \bar{x}\right\rangle\right] = \frac{1}{n^2} \sum_{i \in [n]} \sum_{j \neq i} \mathbb{E}\left[\frac{1}{(A^{(i)} + 1)(A^{(j)} + 1)}\right]\left\langle x^{(i)} - \bar{x}, \, x^{(j)} - \bar{x}\right\rangle
$$

$$
+ \frac{2}{n^2} \sum_{i \in [n]} \sum_{j \neq i} \sum_{k \neq i, k \neq j} \mathbb{E}\left[\frac{\mathcal{I}_k^{(i)}}{(A^{(i)} + 1)(A^{(j)} + 1)}\right]\left\langle x^{(k)} - \bar{x}, \, x^{(j)} - \bar{x}\right\rangle
$$

$$
+ \frac{2}{n^2} \sum_{i \in [n]} \sum_{j \neq i} \sum_{k \neq i, k \neq j} \sum_{l \neq i, l \neq j, l \neq k} \mathbb{E}\left[\frac{\mathcal{I}_k^{(i)} \mathcal{I}_l^{(j)}}{(A^{(i)} + 1)(A^{(j)} + 1)}\right]\left\langle x^{(k)} - \bar{x}, \, x^{(l)} - \bar{x}\right\rangle \quad (27)
$$

Now note that by symmetry, for any $i, j \in [n]$, we have

$$
\mathbb{E}\left[\frac{1}{(A^{(i)} + 1)(A^{(j)} + 1)}\right] = \mathbb{E}\left[\frac{1}{(A^{(1)} + 1)(A^{(2)} + 1)}\right].
$$

Similarly,

$$
\mathbb{E}\left[\frac{\mathcal{I}_k^{(i)}}{(A^{(i)} + 1)(A^{(j)} + 1)}\right] = \mathbb{E}\left[\frac{\mathcal{I}_3^{(1)}}{(A^{(1)} + 1)(A^{(2)} + 1)}\right],
$$

and

$$
\mathbb{E}\left[\frac{\mathcal{I}_k^{(i)} \mathcal{I}_l^{(j)}}{(A^{(i)} + 1)(A^{(j)} + 1)}\right] = \mathbb{E}\left[\frac{\mathcal{I}_3^{(1)} \mathcal{I}_4^{(2)}}{(A^{(1)} + 1)(A^{(2)} + 1)}\right].
$$

This implies that all three terms in (27) can be written as

$$
c \sum_{i \in [n]} \sum_{j \neq i} \left\langle x^{(i)} - \bar{x}, \, x^{(j)} - \bar{x}\right\rangle,
$$

where $c$ is a positive constant. We also have

$$
\sum_{i \in [n]} \sum_{j \neq i} \left\langle x^{(i)} - \bar{x}, \, x^{(j)} - \bar{x}\right\rangle = \sum_{i \in [n]} \left\langle x^{(i)} - \bar{x}, \, \sum_{j \neq i}(x^{(j)} - \bar{x})\right\rangle
$$

$$
= -\sum_{i \in [n]} \left\|x^{(i)} - \bar{x}\right\|^2.
$$

Therefore all the terms in (27) are non-positive. Combining this with (26), we obtain that

$$\mathbb{E}\left[\left\|\bar{y} - \bar{x}\right\|^2\right] \leq \frac{1}{n^2} \sum_{i \in [n]} \mathbb{E}\left[\left\|y^{(i)} - \bar{x}\right\|^2\right]$$

$$\leq \frac{\beta_s}{n} \cdot \frac{1}{n} \sum_{i \in [n]} \left\|x^{(i)} - \bar{x}\right\|^2,$$

where the second inequality uses (25). Combining this with Lemma 5 then concludes the proof. □

**Remark 2.** *Note that $\beta_s$ computed in Lemma 1 is decreasing in $s$ and increasing in $n$, therefore, for any $s \geq 1$, and $n \geq 2$, we have*

$$\beta_s\bigg|_{n<\infty} \leq \lim_{n\to\infty} \beta_1$$

$$= \lim_{n\to\infty} \left(1 - \left(1 - \frac{1}{n-1}\right)^n - \frac{1}{n-1}\right)$$

$$= 1 - \frac{1}{e},$$

*where $e$ is Euler's Number and we used the fact that $\lim_{n\to\infty}\left(1 - \frac{1}{n}\right)^n = \frac{1}{e}$. Similarly, for any $s \geq 1$, and $n \geq 2$, we have*

$$\alpha_s\bigg|_{n<\infty} \leq \lim_{n\to\infty} \alpha_1 = \frac{1}{2} < 1 - \frac{1}{e}.$$

## B.2 Proof of Lemma 2

**Lemma 2.** *Suppose that assumptions 1, 2, and 3 hold true. Consider Algorithm 1. Consider a step-size $\gamma$ such that $\gamma \leq \frac{1}{20L}$. For any $t \geq 0$, we obtain that*

$$\frac{1}{n^2} \sum_{i,j\in[n]} \mathbb{E}\left[\left\|x_t^{(i)} - x_t^{(j)}\right\|^2\right] \leq 20\frac{1 + 3\eta_s}{(1 - \eta_s)^2}\eta_s\gamma^2\left(\sigma^2 + \mathcal{H}^2\right),$$

*and*

$$\frac{1}{n^2} \sum_{i,j\in[n]} \mathbb{E}\left[\left\|g_t^{(i)} - g_t^{(j)}\right\|^2\right] \leq 15\left(\sigma^2 + \mathcal{H}^2\right),$$

*where $\eta_s = \alpha_s$ for EL-Oracle and $\eta_s = \beta_s$ for EL-Local as defined in Lemma 1.*

*Proof.* First note that for any $i \in [n]$, we have

$$g_t^{(i)} - g_t^{(j)} = g_t^{(i)} - \nabla f^{(i)}\left(x_t^{(i)}\right) + \nabla f^{(i)}\left(x_t^{(i)}\right) - \nabla f^{(i)}\left(\bar{x}_t\right) + \nabla f^{(i)}\left(\bar{x}_t\right)$$

$$- \nabla f^{(j)}\left(\bar{x}_t\right) + \nabla f^{(j)}\left(\bar{x}_t\right) - \nabla f^{(j)}\left(x_t^{(j)}\right) + \nabla f^{(j)}\left(x_t^{(j)}\right) - g_t^{(j)}$$

Thus, using Lemma 4, we have

$$\left\|g_t^{(i)} - g_t^{(j)}\right\|^2 \leq 5\left\|g_t^{(i)} - \nabla f^{(i)}\left(x_t^{(i)}\right)\right\|^2 + 5\left\|\nabla f^{(i)}\left(x_t^{(i)}\right) - \nabla f^{(i)}\left(\bar{x}_t\right)\right\|^2$$

$$+ 5\left\|\nabla f^{(j)}\left(x_t^{(j)}\right) - \nabla f^{(j)}\left(\bar{x}_t\right)\right\|^2 + 5\left\|g_t^{(j)} - \nabla f^{(j)}\left(x_t^{(j)}\right)\right\|^2$$

$$+ 5\left\|\nabla f^{(i)}\left(\bar{x}_t\right) - \nabla f^{(j)}\left(\bar{x}_t\right)\right\|^2.$$

Taking the conditional expectation, we have

$$\mathbb{E}_t\left[\left\|g_t^{(i)} - g_t^{(j)}\right\|^2\right] \leq 5\mathbb{E}_t\left[\left\|g_t^{(i)} - \nabla f^{(i)}\left(x_t^{(i)}\right)\right\|^2\right] + 5\mathbb{E}_t\left[\left\|\nabla f^{(i)}\left(x_t^{(i)}\right) - \nabla f^{(i)}\left(\bar{x}_t\right)\right\|^2\right]$$

$$+ 5\mathbb{E}_t\left[\left\|\nabla f^{(j)}\left(x_t^{(j)}\right) - \nabla f^{(j)}\left(\bar{x}_t\right)\right\|^2\right] + 5\mathbb{E}_t\left[\left\|g_t^{(j)} - \nabla f^{(j)}\left(x_t^{(j)}\right)\right\|^2\right]$$

$$+ 5\mathbb{E}_t\left[\left\|\nabla f^{(i)}\left(\bar{x}_t\right) - \nabla f^{(j)}\left(\bar{x}_t\right)\right\|^2\right] \tag{28}$$

Now by Assumption 2, we have

$$\mathbb{E}_t\left[\left\|g_t^{(i)} - \nabla f^{(i)}\left(x_t^{(i)}\right)\right\|^2\right] \leq \sigma^2. \tag{29}$$

By Assumption 1, we have

$$\mathbb{E}_t\left[\left\|\nabla f^{(i)}\left(x_t^{(i)}\right) - \nabla f^{(i)}\left(\bar{x}_t\right)\right\|^2\right] \leq L^2 \mathbb{E}_t\left[\left\|x_t^{(i)} - \bar{x}_t\right\|^2\right]. \tag{30}$$

Thus, by Assumption 3, and Lemma 5, we obtain that

$$\frac{1}{n^2}\sum_{i,j\in[n]}\mathbb{E}_t\left[\left\|\nabla f^{(i)}\left(\bar{x}_t\right) - \nabla f^{(j)}\left(\bar{x}_t\right)\right\|^2\right] \leq 2\mathcal{H}^2. \tag{31}$$

Combining (28), (29), (30), and (31), and taking total expectation from both sides, we obtain that

$$\frac{1}{n^2}\sum_{i,j\in[n]}\mathbb{E}\left[\left\|g_t^{(i)} - g_t^{(j)}\right\|^2\right] \leq \frac{10L^2}{n}\sum_{i\in[n]}\mathbb{E}\left[\left\|x_t^{(i)} - \bar{x}_t\right\|^2\right] + 10\sigma^2 + 10\mathcal{H}^2. \tag{32}$$

Now Lemma 5 yields

$$\frac{1}{n^2}\sum_{i,j\in[n]}\mathbb{E}\left[\left\|g_t^{(i)} - g_t^{(j)}\right\|^2\right] \leq \frac{5L^2}{n^2}\sum_{i,j\in[n]}\mathbb{E}\left[\left\|x_t^{(i)} - x_t^{(j)}\right\|^2\right] + 10\sigma^2 + 10\mathcal{H}^2. \tag{33}$$

We now analyze $\frac{1}{n^2}\sum_{i,j\in[n]}\mathbb{E}\left[\left\|x_t^{(i)} - x_t^{(j)}\right\|^2\right]$. From Algorithm 1, recall that for all $i \in [n]$, we have $x_{t+1/2}^{(i)} = x_t^{(i)} - \gamma g_t^{(i)}$. We obtain for all $i, j \in [n]$, that

$$\mathbb{E}\left[\left\|x_{t+1/2}^{(i)} - x_{t+1/2}^{(j)}\right\|^2\right] \leq \mathbb{E}\left[\left\|x_t^{(i)} - x_t^{(j)} - \gamma\left(g_t^{(i)} - g_t^{(j)}\right)\right\|^2\right] \tag{34}$$

$$\leq (1+c)\mathbb{E}\left[\left\|x_t^{(i)} - x_t^{(j)}\right\|^2\right] + \left(1 + \frac{1}{c}\right)\gamma^2\mathbb{E}\left[\left\|g_t^{(i)} - g_t^{(j)}\right\|^2\right],$$

where we used the fact that $(x+y)^2 \leq (1+c)x^2 + (1+1/c)y^2$ for any $c > 0$. Now recall that $\eta_s = \alpha_s$ for EL-Oracle, and $\eta_s = \beta_s$ for EL-Local. Therefore, by Property $(b)$ of Lemma 1, for both variant of the algorithm, we have

$$\frac{1}{n}\sum_{i\in[n]}\mathbb{E}\left[\left\|x_{t+1}^{(i)} - \bar{x}_{t+1}\right\|^2\right] \leq \frac{\eta_s}{n}\sum_{i\in[n]}\mathbb{E}\left[\left\|x_{t+1/2}^{(i)} - \bar{x}_{t+1/2}\right\|^2\right]$$

Then, applying Lemma 5, we have

$$\frac{1}{n^2}\sum_{i,j\in[n]}\mathbb{E}\left[\left\|x_{t+1}^{(i)} - x_{t+1}^{(j)}\right\|^2\right] \leq \frac{\eta_s}{n^2}\sum_{i,j\in[n]}\mathbb{E}\left[\left\|x_{t+1/2}^{(i)} - x_{t+1/2}^{(j)}\right\|^2\right].$$

Combining this with (34), we obtain that

$$\frac{1}{n^2}\sum_{i,j\in[n]}\mathbb{E}\left[\left\|x_{t+1}^{(i)} - x_{t+1}^{(j)}\right\|^2\right] \leq (1+c)\eta_s\frac{1}{n^2}\sum_{i,j\in[n]}\mathbb{E}\left[\left\|x_t^{(i)} - x_t^{(j)}\right\|^2\right]$$

$$+ \left(1 + \frac{1}{c}\right)\eta_s\gamma^2\frac{1}{n^2}\sum_{i,j\in[n]}\mathbb{E}\left[\left\|g_t^{(i)} - g_t^{(j)}\right\|^2\right].$$

For $c = \frac{1-\eta_s}{4\eta_s}$, we obtain that

$$\frac{1}{n^2}\sum_{i,j\in[n]}\mathbb{E}\left[\left\|x_{t+1}^{(i)} - x_{t+1}^{(j)}\right\|^2\right] \leq \frac{1+3\eta_s}{4}\frac{1}{n^2}\sum_{i,j\in[n]}\mathbb{E}\left[\left\|x_t^{(i)} - x_t^{(j)}\right\|^2\right]$$

$$+ \frac{1+3\eta_s}{1-\eta_s}\eta_s\gamma^2\frac{1}{n^2}\sum_{i,j\in[n]}\mathbb{E}\left[\left\|g_t^{(i)} - g_t^{(j)}\right\|^2\right].$$

Combining this with (33), we obtain that

$$
\frac{1}{n^2} \sum_{i,j \in [n]} \mathbb{E} \left[ \left\| x_{t+1}^{(i)} - x_{t+1}^{(j)} \right\|^2 \right] \leq \frac{1 + 3\eta_s}{4} \frac{1}{n^2} \sum_{i,j \in [n]} \mathbb{E} \left[ \left\| x_t^{(i)} - x_t^{(j)} \right\|^2 \right]
$$

$$
+ \frac{1 + 3\eta_s}{1 - \eta_s} \eta_s \gamma^2 \left( \frac{5L^2}{n^2} \sum_{i,j \in [n]} \mathbb{E} \left[ \left\| x_t^{(i)} - x_t^{(j)} \right\|^2 \right] + 10\sigma^2 + 10\mathcal{H}^2 \right)
$$

$$
= \left( \frac{1 + 3\eta_s}{4} + 5 \frac{1 + 3\eta_s}{1 - \eta_s} \eta_s \gamma^2 L^2 \right) \frac{1}{n^2} \sum_{i,j \in [n]} \mathbb{E} \left[ \left\| x_t^{(i)} - x_t^{(j)} \right\|^2 \right]
$$

$$
+ \frac{1 + 3\eta_s}{1 - \eta_s} \eta_s \gamma^2 \left( 10\sigma^2 + 10\mathcal{H}^2 \right).
$$

Now note that for by Remark 2, for both variants of the algorithm, we have $\eta_s \leq 1 - \frac{1}{e}$, which implies that

$$
\gamma^2 \leq \frac{1}{(20L)^2} \leq \frac{(1 - \eta_s)^2}{20\eta_s(1 + 3\eta_s)L^2}.
$$

Therefore,

$$
\frac{1}{n^2} \sum_{i,j \in [n]} \mathbb{E} \left[ \left\| x_{t+1}^{(i)} - x_{t+1}^{(j)} \right\|^2 \right] \leq \frac{1 + \eta_s}{2} \frac{1}{n^2} \sum_{i,j \in [n]} \mathbb{E} \left[ \left\| x_t^{(i)} - x_t^{(j)} \right\|^2 \right] + \frac{1 + 3\eta_s}{1 - \eta_s} \eta_s \gamma^2 \left( 10\sigma^2 + 10\mathcal{H}^2 \right).
$$

Unrolling the recursion, we obtain that

$$
\frac{1}{n^2} \sum_{i,j \in [n]} \mathbb{E} \left[ \left\| x_t^{(i)} - x_t^{(j)} \right\|^2 \right] \leq 20 \frac{1 + 3\eta_s}{(1 - \eta_s)^2} \eta_s \gamma^2 \left( \sigma^2 + \mathcal{H}^2 \right).
$$

Combining this with (32), we obtain that

$$
\frac{1}{n^2} \sum_{i,j \in [n]} \mathbb{E} \left[ \left\| g_t^{(i)} - g_t^{(j)} \right\|^2 \right] \leq 15 \left( \sigma^2 + \mathcal{H}^2 \right).
$$

This is the desired result. $\qquad \square$

### B.3  Proof of Lemma 3

**Lemma 3.** *Suppose that assumptions 1 and 2 hold true. Consider Algorithm 1 with $\gamma \leq \frac{1}{2L}$. For any $t \in \{0, \ldots, T - 1\}$, we obtain that*

$$
\mathbb{E} \left[ \left\| \nabla F \left( \bar{x}_t \right) \right\|^2 \right] \leq \frac{2}{\gamma} \mathbb{E} \left[ F(\bar{x}_t) - F(\bar{x}_{t+1}) \right] + \frac{L^2}{2n^2} \sum_{i,j \in [n]} \mathbb{E} \left[ \left\| x_t^{(i)} - x_t^{(j)} \right\|^2 \right]
$$

$$
+ 2L\gamma \frac{\sigma^2}{n} + \frac{2L}{\gamma} \mathbb{E} \left[ \left\| \bar{x}_{t+1} - \bar{x}_{t+1/2} \right\|^2 \right].
$$

*Proof.* Consider an arbitrary $t \in [T]$. Note that, we do not necessarily have $\bar{x}_{t+1} = \bar{x}_{t+1/2}$. But by Lemma 1, we have $\mathbb{E} \left[ \bar{x}_{t+1} \right] = \mathbb{E} \left[ \bar{x}_{t+1/2} \right]$. Now by the smoothness of the loss function (Assumption 1), we have

$$
F(\bar{x}_{t+1}) - F(\bar{x}_t) \leq \langle \bar{x}_{t+1} - \bar{x}_t, \nabla F \left( \bar{x}_t \right) \rangle + \frac{L}{2} \left\| \bar{x}_{t+1} - \bar{x}_t \right\|^2
$$

$$
= \langle \bar{x}_{t+1} - \bar{x}_{t+1/2} + \bar{x}_{t+1/2} - \bar{x}_t, \nabla F \left( \bar{x}_t \right) \rangle + \frac{L}{2} \left\| \bar{x}_{t+1} - \bar{x}_{t+1/2} + \bar{x}_{t+1/2} - \bar{x}_t \right\|^2.
$$

Now denoting $\overline{\nabla F}_t = \frac{1}{n}\sum_{i\in[n]}\nabla f^{(i)}\left(x_t^{(i)}\right)$, $\bar{g}_t = \frac{1}{n}\sum_{i\in[n]}g_t^{(i)}$ and taking the conditional expectation, we have

$$\mathbb{E}_t\left[F(\bar{x}_{t+1})\right] - F(\bar{x}_t) \leq \left\langle \mathbb{E}_t\left[\bar{x}_{t+1/2} - \bar{x}_t\right], \nabla F\left(\bar{x}_t\right)\right\rangle + \frac{L}{2}\mathbb{E}_t\left[\left\|\bar{x}_{t+1} - \bar{x}_{t+1/2} + \bar{x}_{t+1/2} - \bar{x}_t\right\|^2\right]$$

$$\overset{(a)}{\leq} -\gamma\left\langle\overline{\nabla F}_t, \nabla F\left(\bar{x}_t\right)\right\rangle + L\gamma^2\mathbb{E}_t\left[\left\|\bar{g}_t\right\|^2\right] + L\mathbb{E}_t\left[\left\|\bar{x}_{t+1} - \bar{x}_{t+1/2}\right\|^2\right]$$

$$\overset{(b)}{\leq} -\gamma\left\langle\overline{\nabla F}_t, \nabla F\left(\bar{x}_t\right)\right\rangle + L\gamma^2\mathbb{E}_t\left[\left\|\overline{\nabla F}_t\right\|^2\right] + L\gamma^2\frac{\sigma^2}{n} + L\mathbb{E}_t\left[\left\|\bar{x}_{t+1} - \bar{x}_{t+1/2}\right\|^2\right].$$
(35)

where (a) uses Young's inequality and (b) is based on the facts that by Assumption 2, we have $\mathbb{E}_t\left[\bar{g}_t\right] = \overline{\nabla F}_t$, and $\mathbb{E}_t\left[\left\|\bar{g}_t - \overline{\nabla F}_t\right\|^2\right] \leq \frac{\sigma^2}{n}$. Now note that using the fact that $\gamma \leq \frac{1}{2L}$, we have

$$-\gamma\left\langle\overline{\nabla F}_t, \nabla F\left(\bar{x}_t\right)\right\rangle + L\gamma^2\left\|\overline{\nabla F}_t\right\|^2 \leq \frac{\gamma}{2}\left(-2\left\langle\overline{\nabla F}_t, \nabla F\left(\bar{x}_t\right)\right\rangle + \left\|\overline{\nabla F}_t\right\|^2\right)$$

$$= \frac{\gamma}{2}\left(-\left\|\nabla F\left(\bar{x}_t\right)\right\|^2 + \left\|\nabla F\left(\bar{x}_t\right) - \overline{\nabla F}_t\right\|^2\right). \quad (36)$$

Combining (35), and (36), we obtain that

$$\mathbb{E}_t\left[F(\bar{x}_{t+1})\right] - F(\bar{x}_t) \leq -\frac{\gamma}{2}\left\|\nabla F\left(\bar{x}_t\right)\right\|^2 + \frac{\gamma}{2}\left\|\nabla F\left(\bar{x}_t\right) - \overline{\nabla F}_t\right\|^2 + L\gamma^2\frac{\sigma^2}{n} + L\mathbb{E}_t\left[\left\|\bar{x}_{t+1} - \bar{x}_{t+1/2}\right\|^2\right].$$

Taking total expectation, we obtain that

$$\mathbb{E}\left[F(\bar{x}_{t+1}) - F(\bar{x}_t)\right] \leq -\frac{\gamma}{2}\mathbb{E}\left[\left\|\nabla F\left(\bar{x}_t\right)\right\|^2\right] + \frac{\gamma}{2}\mathbb{E}\left[\left\|\nabla F\left(\bar{x}_t\right) - \overline{\nabla F}_t\right\|^2\right]$$

$$+ L\gamma^2\frac{\sigma^2}{n} + L\mathbb{E}\left[\left\|\bar{x}_{t+1} - \bar{x}_{t+1/2}\right\|^2\right]. \quad (37)$$

Now, note that

$$\mathbb{E}\left[\left\|\overline{\nabla F}_t - \nabla F\left(\bar{x}_t\right)\right\|^2\right] = \mathbb{E}\left[\left\|\frac{1}{n}\sum_{i\in[n]}\nabla f^{(i)}(x_t^{(i)}) - \frac{1}{n}\sum_{i\in[n]}\nabla f^{(i)}\left(\bar{x}_t\right)\right\|^2\right]$$

$$= \mathbb{E}\left[\left\|\frac{1}{n}\sum_{i\in[n]}\left(\nabla f^{(i)}(x_t^{(i)}) - \nabla f^{(i)}\left(\bar{x}_t\right)\right)\right\|^2\right]$$

$$\leq \frac{1}{n}\sum_{i\in[n]}\mathbb{E}\left[\left\|\nabla f^{(i)}(x_t^{(i)}) - \nabla f^{(i)}\left(\bar{x}_t\right)\right\|^2\right]$$

$$\overset{(a)}{\leq} \frac{L^2}{n}\sum_{i\in[n]}\mathbb{E}\left[\left\|x_t^{(i)} - \bar{x}_t\right\|^2\right].$$

$$\overset{(b)}{\leq} \frac{L^2}{2n^2}\sum_{i,j\in[n]}\mathbb{E}\left[\left\|x_t^{(i)} - x_t^{(j)}\right\|^2\right],$$

where (a) uses Assumption 1, in (b), we used Lemma 5. Combining this with (37), we obtain that

$$\mathbb{E}\left[F(\bar{x}_{t+1}) - F(\bar{x}_t)\right] \leq -\frac{\gamma}{2}\mathbb{E}\left[\left\|\nabla F\left(\bar{x}_t\right)\right\|^2\right] + \frac{\gamma}{2}\frac{L^2}{2n^2}\sum_{i,j\in[n]}\mathbb{E}\left[\left\|x_t^{(i)} - x_t^{(j)}\right\|^2\right]$$

$$+ L\gamma^2\frac{\sigma^2}{n} + L\mathbb{E}\left[\left\|\bar{x}_{t+1} - \bar{x}_{t+1/2}\right\|^2\right].$$

Rearranging the terms we have

$$\mathbb{E}\left[\left\|\nabla F\left(\bar{x}_t\right)\right\|^2\right] \leq \frac{2}{\gamma}\mathbb{E}\left[F(\bar{x}_t) - F(\bar{x}_{t+1})\right] + \frac{L^2}{2n^2}\sum_{i,j\in[n]}\mathbb{E}\left[\left\|x_t^{(i)} - x_t^{(j)}\right\|^2\right]$$

$$+ 2L\gamma\frac{\sigma^2}{n} + \frac{2L}{\gamma}\mathbb{E}\left[\left\|\bar{x}_{t+1} - \bar{x}_{t+1/2}\right\|^2\right].$$

Table 3: Summary of learning parameters used for evaluation. For each dataset, batch size (b), and the number of training steps per communication round (r) are presented.

| TASK | DATASET | MODEL | b | r | TRAINING SAMPLES | TOTAL PARAMETERS |
|------|---------|-------|---|---|------------------|------------------|
| Image Classification | CIFAR-10 | GN-LeNet | 8 | 3 | 50 000 | 89 834 |
| Image Classification | FEMNIST | CNN | 16 | 7 | 734 463 | 1 690 046 |

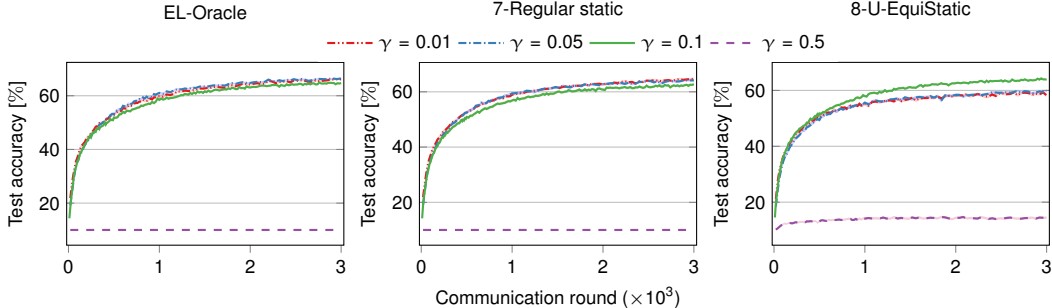

Figure 6: The test accuracy for EL-Oracle, `7-Regular static` topologies, and `8-U-EquiStatic` topologies with different learning rates $\gamma$, with the CIFAR-10 dataset.

which is the lemma.

$\square$

## C Experimental Details and Further Evaluation

### C.1 Experiment setup and learning rate tuning

Table 3 provides a summary of the learning parameters, model, and dataset used in the experiments. The models were evaluated on samples from the test set that were unseen during training. The step size ($\gamma$) was tuned by running each baseline on a range of values and taking the setup with the best validation accuracy. The plots in Figure 6 show the CIFAR-10 convergence plots for EL-Oracle, `7-Regular static`, and `8-U-EquiStatic` (most prominent topologies) over different step sizes. The optimal step-sizes are $\gamma = 0.1$ for `Fully connected` and `8-U-EquiStatic`, and $\gamma = 0.05$ for the remaining algorithms and topologies over CIFAR-10. For FEMNIST, the optimal step size is $\gamma = 0.1$ for all algorithms and topologies. All experiments in Section 4 were conducted for a total of 3000 communication rounds and the experiments for homogeneous data distributions of CIFAR-10 (Figure 7) were conducted for a total of 1000 communication rounds. The experiments over the FEMNIST dataset were conducted for a total of 1440 communication rounds.

### C.2 Computational resources

We perform experiments on 6 hyperthreading-enabled machines with dual Intel(R) Xeon(R) CPU E5-2630 v3 @ 2.40GHz of 8 cores. All algorithms were implemented as part of Decentralizepy [11]. Each DL node was assigned 2 virtual cores. An experiment took at most 3 hours in wall-clock time for experiments in Section 4, 2 hours in wall-clock time for FEMNIST experiments, and 1 hour in wall-clock time for CIFAR-10 experiments in Appendix C, summing to a total of 1152 virtual CPU hours. Across all experiments that are presented in this article, including different seeds and learning rate tuning, the total virtual CPU time goes up to approximately 54 700 hours.

### C.3 Experiments with homogeneous data distributions

In this experiment, we focused on evaluating the performance of different communication topologies under IID data. The data was generated using a Dirichlet distribution with $\alpha = 1.0$. As a reference

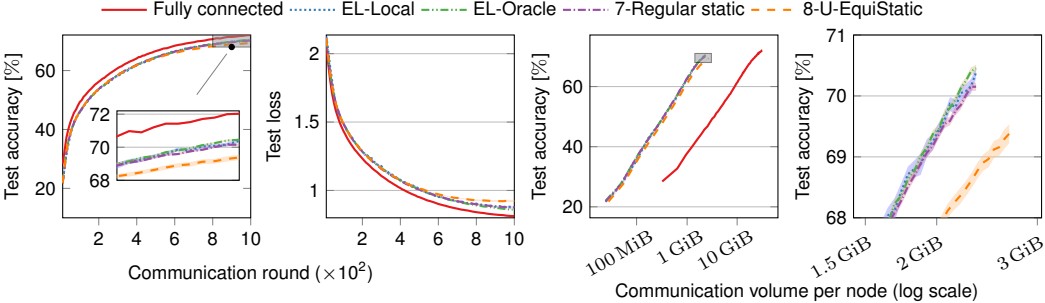

Figure 7: Convergence and communication usage results for homogeneous data partitioning (Dirichlet distribution $\alpha = 1.0$) of the CIFAR-10 dataset.

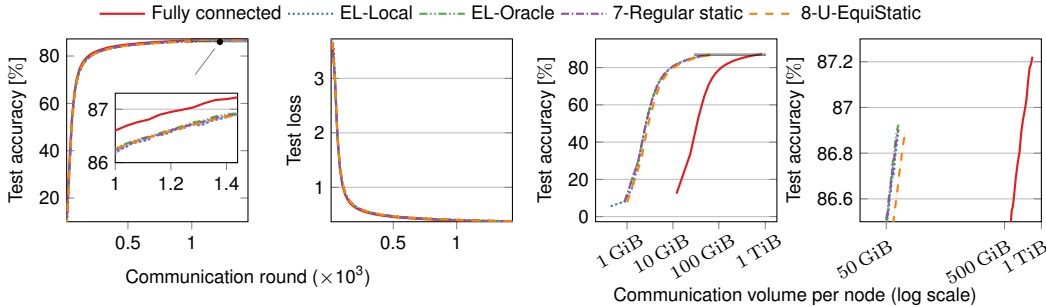

Figure 8: Communication rounds vs. top-1 test accuracy and (left) and communication volume per node vs. test accuracy (right) for the FEMNIST dataset.

baseline, we established a `Fully connected` topology. Additionally, we examined EL (EL-Oracle and EL-Local) in comparison to `7-Regular static` and `8-U-EquiStatic`.

We observed that the results were not particularly exciting (Figure 7), primarily due to the nature of the data distribution, which leaves limited room for improvement across the different topologies. In terms of accuracy, the `Fully connected` topology achieved roughly $2\%$ better accuracy than the sparser topologies. EL-Oracle displayed a marginal advantage over the other approaches. However, the convergence behavior for EL-Local, `7-Regular static`, and EL-Oracle was nearly overlapping, both in terms of the number of rounds and communication usage. On the other hand, the `8-U-EquiStatic` approach lagged behind, exhibiting $1\%$ lower accuracy.

Although significant improvements were not observed in IID data settings, it is worth noting that EL did not hinder convergence. In more realistic non-IID data distributions, EL demonstrated improved accuracy and reduced the network utilization, as shown in Figure 4, Section 4.

### C.4 Performance of EL and baselines on FEMNIST

As an additional dataset, we employed FEMNIST [7] to evaluate EL against D-PSGD over other topologies. Figure 8 shows the convergence plots for the baselines against both EL-Oracle and EL-Local. We observe the same trend with `Fully connected` topology performing the best in terms of achieved top-1 accuracy. EL-Oracle outperforms EL-Local and both achieve higher top-1 accuracy compared to the static topologies `7-Regular static` and `8-U-EquiStatic`. The improvement in accuracy with EL ($0.2\%$ between EL-Oracle and `7-Regular static`) does not look as evident as with the CIFAR-10 dataset because the room for improvement (difference between `7-Regular static` and `Fully connected`) is quite small. This can be attributed to the homogeneity of the FEMNIST dataset and the complexity of the digit recognition task.

## D  Notes on Network Connectivity and EL Performance

The implementation of our decentralized scheme is built around the condition that all nodes can communicate with each other. This is similar to the assumptions of the EquiTopo topologies, a

competitor baseline [53]. However, we argue that the connectivity requirement is a bit more lenient, allowing our EL approach to function in a wide range of practical scenarios. In data center settings, it is common to train on clusters of highly interconnected GPUs, and all-to-all communication should be achievable in these settings. In edge settings, *e.g.*, a network of mobile devices collaboratively training a model while keeping private datasets, the communication barrier might appear more substantial. Nonetheless, Internet networks are generally well-connected, which mitigates this concern. More importantly, from a practical point of view, even if pairwise communications encounter some barriers, the decentralized and randomized nature of EL-Oracle and EL-Local should still allow for effective model learning and convergence. The occasional lack of communication between specific nodes should not significantly impact the algorithm's performance, as model updates are still propagated through other communicating nodes, as long as the network is not partitioned.

EL is most useful in scenarios where every pair of nodes can communicate, but the total communication budget is limited. Our randomized communication scheme allows for efficient use of the limited resources while ensuring faster model convergence than conventional decentralized learning approaches.

