# OpenReview forum: "Epidemic Learning: Boosting Decentralized Learning with Randomized Communication"
_NeurIPS.cc/2023/Conference — NeurIPS 2023 poster_

### Official Review · Reviewer_Grsp · 2023-07-04

**Soundness:** 3 good
**Presentation:** 3 good
**Contribution:** 3 good
**Rating:** 7
**Confidence:** 3

**Summary:**

The authors study the benefits of using randomized communication topologies for decentralized optimization of non-convex functions. Critically, the communication protocols studied are not based on picking/sampling a communication graph that remains fixed throughout learning. Instead the authors study the case where the nodes randomize their communication patterns with the rest of the nodes in each round. Based on this premise they propose two algorithms, EL Oracle and EL local and study their convergence rate. The prosed algorithms are shown to converge asymptotically faster than state of the art. Experiments verify the theoretical advantage empirically.

**Strengths:**

The paper communicates clearly and precisely its contribution. Despite extensive work on decentralized algorithms, including randomized and fixed topologies that can vary over time, to the best of my knowledge the proposed algorithms are novel. Importantly, the authors note that time varying randomized topologies have not been shown to have an advantage over their static counterparts.

**Weaknesses:**

I am not 100% sure which quantity is bounded in (3) and if it is supposed to be comparable to the rates provided in Theorem 1. The paragraphs after (3) list various ways EL Oracle and Local improve over (3) but I cannot say I follow them. For example, the first term in EL Oracle in Theorem 1 and (3) are identical but the comments mention an advantage in the first term. Clarifying this would help.

In its current state, the paper is not adequately explaining which part of the analysis unlocks the potential of non-static randomized communication graphs. I think highlighting which parts of the analysis are critical would help. Intuitively, I would expect that improving communication would have an effect similar to decreasing the variance $\sigma$ or improving $p$ but it seems like this is not true.

**Questions:**

Another thing that was unclear to me is how is this improvement affected by the fact that we are averaging iterates $x$ instead of gradients. Does the advantage persist in this case? In the case where we are averaging gradients I have the intuition that it would be harder to show an advantage but I am not sure.

**Limitations:**

No limitations to address.

---

> ### Author Rebuttal · Authors · 2023-08-09
>
> We thank the reviewer for their comments and provide a detailed rebuttal below.
> ___
> > Q: Which quantity is bounded in (3)? The first term in EL Oracle in Theorem 1 and (3) are identical but the comments mention an advantage in the first term. Clarifying this would help.
>
> Re: The quantity bounded in (3) is the average of the norm of the gradient which is exactly the same as Theorem 1 and (3) should be compared with bounds provided in Theorem 1. However, there seems to be a small misunderstanding regarding our comparison. We do not claim an advantage in the first term. In line 212, we mention that our algorithm preserves linear speed up which is the same as the bound in (3). In fact, it is not possible to outperform this term as it is the same term even for running a centralized SGD on a single node. Preserving linear speed-up may be surprising for EL-Local as in this case, the maxing matrix is not doubly stochastic.
> We claim our bound to have a better second error term which is the main error term when comparing different decentralized learning schemes. Please see our explanation regarding "Transient Iterations" in the global response at the top of the page.
> ___
> > Q: Which part of the analysis unlocks the potential of non-static randomized communication graphs?
>
> Re: To demonstrate the superior convergence of our EL approach, our analysis proceeds in two main steps.
>
> 1) The first step is established in Lemma 1, where we demonstrate that the randomized communication within EL achieves the property of fast mixing, with a mixing coefficient ($\alpha_s$ or $\beta_s$) of $\mathcal{O}(\frac{1}{s})$.
>
> 2) The second step was tightening the conventional analysis of decentralized learning. Let $\beta = 1 - p \in [0,1)$ be the mixing coefficient (a.k.a. consensus rate [23]). Existing analyses of decentralized learning propose a convergence rate (as per Equation (3)) of:
> $$\mathcal{O}\left(\sqrt{\frac{L \Delta_0 \sigma^2}{n T}}+\sqrt[3]{\frac{L^2 \Delta_0^2\sigma^2}{(1-\beta) T^2}}+\sqrt[3]{\frac{L^2 \Delta_0^2\mathcal{H}^2}{(1-\beta)^2 T^2}}+\frac{L \Delta_0}{(1-\beta) T}\right).$$
>
> Here, $\beta$'s impact on the convergence rate only appears in terms formatted as $\mathcal{O}\left(\frac{1}{(1-\beta)^k}\right)$. Therefore, as long as $\frac{1}{1-\beta} \in \mathcal{O}(1)$, diminishing $\beta$ does not affect the order of the convergence rate. Hence, e.g., $\beta = \frac{1}{2}$ and $\beta = \frac{1}{100}$ would have identical convergence rates (with both yielding $\frac{1}{1-\beta} \in \mathcal{O}(1)$ and the exact value of $\frac{1}{1-\beta}$ varying by a factor of less than 2). This suggests that we cannot achieve a convergence rate better than the existing state-of-the-art solutions with $\frac{1}{1-\beta} \in \mathcal{O}(1)$. However, we observed that this limitation arises from a looseness in the conventional analysis. More specifically, the second and third terms in the above expression can be tightened such that they go to zero as $\beta$ approaches zero (see Theorem 1, and the discussion below the theorem).  This adjustment allows us to showcase the benefits of EL, which maintains a $\beta$ in $\mathcal{O}(\frac{1}{s})$.
> ___
>
> > Q: What happens if we average gradients instead of models $x$?
>
> Re:  In the context of decentralized learning, unlike federated learning, averaging gradients may lead to divergence among local models of nodes. As a result, both our work and the existing related works we compared against employ the strategy of averaging the local models.
> We agree that investigating the effects of averaging gradients in the context of decentralized learning could be an interesting avenue for future research. However, it is important to note that doing so necessitates the introduction of additional mechanisms to prevent model drift, given the potential divergence issue.

---

> > ### Comment · Reviewer_Grsp · 2023-08-15
> > **My questions are answered**
> >
> > I would like to thank the authors for their thorough responses. I would suggest adding at least a summary of this discussion in the main paper. I keep my score and acceptance recommendation as is.

---

> > > ### Author Response · Authors · 2023-08-15
> > >
> > > We thank the reviewer for their comments and response, and are glad to hear that the reviewer recommends accepting our work. We will add a discussion to the camera-ready version where we will have an extra page.

---

### Official Review · Reviewer_fwhW · 2023-07-06

**Soundness:** 2 fair
**Presentation:** 3 good
**Contribution:** 2 fair
**Rating:** 5
**Confidence:** 3

**Summary:**

This paper proposes a decentralized learning algorithm based on random communication, i.e., each node sends its model to a random set (with a fixed size) of other nodes at each round. This paper theoretically shows the superiority of random communication in terms of transient iterations over other decentralized algorithms, which is further validated by experiments.

**Strengths:**

1. This paper provides a simple yet effective scheme of communication for decentralized learning.

2. This paper is technically sound. The technique developed for convergence analysis is interesting.

**Weaknesses:**

1. In the proposed algorithm, each node is required to send message to a fixed number $k$ of random neighbors at each round. In practice, this may only be applied to networks with high connectivity (e.g., fully connected), and other application scenarios need to be motivated. In addition, the theoretical superiority of convergence directly depends on $k$, so it might be unfair comparing with other algorithms which conventionally works on arbitrary connected graphs. Can the restriction on $k$ or uniformly random selection be relaxed a little bit, say, allow to communicate with varying number of neighbors?

2. Although the authors conduct an experiment to explain the imbalanced load on different nodes, I still have a concern that the balanced pattern reported in this paper may rely on the uniformly random selection. Such an assumption may not be aligned with the real cases, e.g., the degrees of different nodes vary, then some nodes may have higher load than others under random selection. This problem may arise from the non-doubly stochasticity of the mixing matrices. Could other methods handling asymmetric communication, such as push-sum, be helpful for load balancing?

3. Figure 4 shows that EL outperforms the baseline algorithm throughout the learning process, which seems inconsistent with the theoretical statement that the superiority of convergence lies in transient iterations, i.e., the early stage. As the learning proceeds, the convergence will be dominated by the first term. So I suppose the test accuracy of the various methods may get closer with large $T$?

**Questions:**

See the weaknesses.

**Limitations:**

Limitations have been properly discussed.

---

> ### Author Rebuttal · Authors · 2023-08-09
>
> We thank the reviewer for their comments and provide a detailed rebuttal below.
> ___
>
> > Q: Only applicable to networks with high connectivity. other application scenarios need to be motivated.
>
> Re: Thank you for your insightful comment.
> You correctly identified that the implementation of our decentralized scheme is built around the condition that all nodes can communicate with each other.
> This is similar to the assumptions of the EquiTopo topologies [51], a competitor baseline.
> However, we argue that the connectivity requirement is a bit more lenient, allowing our model to function in a wide range of practical scenarios.
> In data center settings it is common to train on clusters of highly interconnected GPUs, and all-to-all communication should be achievable in these settings.
> In edge settings, e.g., a network of mobile devices collaboratively training a model while keeping private datasets, the communication barrier might appear more substantial.
> Nonetheless, Internet networks are generally well-connected, which mitigates this concern.
> More importantly, from a practical point of view, even if pairwise communications encounter some barriers, the decentralized and randomized nature of EL-Oracle and EL-Local should still allow for effective model learning and convergence.
> The occasional lack of communication between specific nodes should not significantly impact the algorithm's performance, as model updates are still propagated through other communicating nodes, as long as the network is not partitioned.
>
> EL is most useful in scenarios where every pair of nodes can communicate, but the total communication budget is limited.
> Our randomized communication scheme allows for efficient use of the limited resources while ensuring faster model convergence than conventional decentralized learning approaches.
>
> We will incorporate these explanations into the camera-ready version of the paper to clarify the assumptions and practical implications of our work.
> ___
> > Q: The theoretical superiority of convergence directly depends on $k$.
>
> Re: Indeed, the superiority of our convergence rate is directly linked to the number of random neighbors $k$ (or $s$ with the notation used in the paper), providing a clear trade-off between convergence speed and communication complexity. This flexibility is a distinguishing aspect of our approach.
>
> Specifically, we obtain a state-of-the-art convergence rate for $k = 1$, with a number of transient iterations in $\mathcal{O}(n^3)$, and the rates can be further improved by increasing $k$. While there are existing works that allow several communication levels [51, 55], unlike our method, their convergence rate does not enhance by augmenting the communication budget.
>
> To obtain this improved convergence guarantee, we needed to enhance the convergence analysis, as the existing analysis suggested that it would not be possible to further improve the convergence rate over the state-of-the-art results and to obtain a number of transient iterations less than  $\mathcal{O}(n^3)$ even for communication graphs with high connectivity. For a more detailed theoretical discussion on how we addressed this, please refer to our response to the last concern of reviewer f2yu.
> ___
> > Q: Can the restriction on $k$ or uniformly random selection be relaxed, say, allow to communicate with varying number of neighbors?
>
> Re: This is an interesting idea and we can give each node a "personalized" fan-out value based on characteristics of that node, such as bandwidth capabilities or training speed.
> In the current form of our algorithm, setting $k$ differently for different nodes would bias the model to fit better the data distribution of nodes with a higher fanout. To solve that issue, one needs to give more weight to the updates coming from less active nodes.
> Since this requires significant modifications to our analysis, we consider this extension beyond the scope of our work.
>
> It is also worth noting that even though there are some convergence results for arbitrary graphs [23], these results often depend on the spectral gap of the underlying graph, which is a very challenging parameter to compute for an arbitrary graph. As a result most of the theoretical works often rely on graphs with balanced degrees such as ring, torus, grid, exponential and EquiTopo graphs.
> ___
> > Q: Imbalanced communication load:
>
> Re: We note that this imbalance only arises in EL-Local as the number of received models per round in EL-Oracle is the same for all nodes.
> It might indeed happen in EL-Local that some nodes have to process more incoming models during a round than others.
> Yet, this concern can be addressed by having a node refuse an incoming model transfer when it has already received a particular number of models during a round, and having the sender node retry model exchange with another random node that is less occupied.
> We will add these details in the camera-ready version of our work.
>
> In larger networks, where nodes usually do not know about the participation of all other nodes, it is common to use a peer sampling service to randomly select other nodes [15]. A peer sampling service is a primitive that provides each node with an (almost) uniformly random subset of nodes, and this subset is periodically refreshed. Using an appropriate peer sampler will also help in balancing out the load between different nodes.
> ___
> > Q: As the improvement is in the number of transient iterations, the test accuracy of the various methods may get closer with large $T$, which contradicts figure 4.
>
> Re: Less number of "transient iterations" does not merely imply that an algorithm is faster only during the initial stages of learning. Instead, it indicates that the algorithm has a smaller second error term, which in turn causes the algorithm to perform better during the learning procedure.  Please see our explanation regarding "Transient Iterations" in the global response at the top of the page.

---

> > ### Comment · Reviewer_fwhW · 2023-08-20
> >
> > Thank you for the response. My questions regarding the application scenario and the empirical improvement are settled, while the restriction on $k$ or random selection still seems a limitation of this work.
> >
> > As my concerns are partially settled, I will raise my score to 5.

---

### Official Review · Reviewer_f2yu · 2023-07-06

**Soundness:** 3 good
**Presentation:** 3 good
**Contribution:** 3 good
**Rating:** 6
**Confidence:** 1

**Summary:**

This paper considers Epidemic Learning, a framework for distributed optimization where each node in a network pushes gradient-descent updates to a uniform random subset of $s$ nodes in the network. Theoretical bounds on the rate of convergence are derived as well as the number of ``transient iterations," showing that this scheme improves upon existing fixed and randomized topologies. These results are supplemented by some empirical evaluations that show gains in communication and iterations to convergence compared to complete or fixed sparse topologies.

**Strengths:**

The technical results generalize existing convergence rates of complete graphs (i.e. centralized SGD). The analysis is fairly clean and logical, and the paper is fairly well-written. This work also provides some experimental components to support their findings.

**Weaknesses:**

The practical benefits of this kind of decentralized optimization could be better contextualized (see below). The analysis, while clean, does not seem particularly conceptually surprising.


**Questions:**

---As I am not really an expert on these kinds of methods, I'd defer to the other reviewers as to the novelty of this work's approach. I'd be happy to revise my score in light of any such discussions.

---Is there a succinct summary of where the convergence benefits arise compared to, say, fast mixing but fixed network topologies? Intuitively, it seems to be because the convergence is dominated by the second-largest eigenspace, while random averaging avoids any particular bad eigenspace in expectation.

---To elaborate on the above ``Weaknesses," one aspect that was not clear to me (and perhaps could be elaborated more) is the relevant tradeoffs in implementing such a decentralized scheme. More concretely, in what scenarios should this kind of randomized scheme that nonetheless must permit communication between any pair of nodes be employed? If the communication architecture is itself a limiting bottleneck, then this scheme cannot work due to the need for pairwise communications. Is the proposed use case meant primarily as a way to speed up/reduce optimization problems that already employ complete topologies?

---While I am not a systems expert either, are there other tradeoffs one must consider with respect to routing/scheduling these randomized communications? For instance, it seems fairly intuitive how to easily route messages in, say, a ring topology --- when each node now must route to $s$ random nodes and received messages from $\Omega(s)$ random nodes, is there any ``scheduling'' overhead in practice?

---While I did not get the chance to carefully check the analysis in the Supplementary Material, the analysis seemed quite clean and logical, i.e. write out the various expected quantities in the natural way and derive relatively standard-looking GD-type recurrences. Conversely, this could mean that the analysis itself may not lead to new technical insights for others in the community.

---

> ### Author Rebuttal · Authors · 2023-08-09
>
> We thank the reviewer for their comments and provide a detailed rebuttal below. Given that the reviewer has such a good intuition about the problem and our solution, we are a bit surprised by the low confidence score of the review.
> ___
> > Q: summary of where the convergence benefits arise ...
>
> Re: The intuition of the reviewer is indeed correct. The convergence behavior of decentralized learning schemes is indeed closely related to the second eigenvalue, or spectral gap, of the mixing matrix which governs the speed of mixing. In our approach, with random communication and under the same communication budget compared to a static topology, we can obtain very fast mixing properties. Please see our response to your last question for a detailed discussion of the theoretical significance of our work.
> ___
> > Q: in what scenarios should this kind of randomized scheme that nonetheless must permit communication between any pair of nodes be employed?
>
>
> Re: Thank you for your insightful comment.
> You correctly identified that the implementation of our decentralized scheme is built around the condition that all nodes can communicate with each other.
> This is similar to the assumptions of the EquiTopo topologies [51], a competitor baseline.
> However, we argue that the connectivity requirement is a bit more lenient, allowing our model to function in a wide range of practical scenarios.
> In data center settings it is common to train on clusters of highly interconnected GPUs, and all-to-all communication should be achievable in these settings.
> In edge settings, e.g., a network of mobile devices collaboratively training a model while keeping private datasets, the communication barrier might appear more substantial.
> Nonetheless, Internet networks are generally well-connected, which mitigates this concern.
> More importantly, from a practical point of view, even if pairwise communications encounter some barriers, the decentralized and randomized nature of EL-Oracle and EL-Local should still allow for effective model learning and convergence.
> The occasional lack of communication between specific nodes should not significantly impact the algorithm's performance, as model updates are still propagated through other communicating nodes, as long as the network is not partitioned.
> EL is most useful in scenarios where every pair of nodes can communicate, but the total communication budget is limited.
> Our randomized communication scheme allows for efficient use of the limited resources while ensuring faster model convergence than conventional decentralized learning approaches.
> We will incorporate these explanations into the camera-ready version of the paper to clarify the assumptions and practical implications of our work.
> ___
> > Q: ``scheduling'' overhead in practice?
>
> Re: The topology construction in EL-Oracle and the peer sampler in EL-Local might introduce some communication and computation overhead, but this overhead is absolutely minimal compared to the resources used for model exchange and training.
> ___
> > Q: technical insights of the analysis for others in the community:
>
> Re:
> We are happy that the reviewer finds our analysis clean and logical.
> To demonstrate the superior convergence of our EL approach, our analysis proceeds in two main steps.
>
> 1) The first step is established in Lemma 1, where we demonstrate that the randomized communication within EL achieves the property of fast mixing, with a mixing coefficient ($\alpha_s$ or $\beta_s$) of $\mathcal{O}(\frac{1}{s})$.
>
> 2) The second step was tightening the conventional analysis of decentralized learning. Let $\beta = 1 - p \in [0,1)$ be the mixing coefficient (a.k.a. consensus rate [23]). Existing analyses of decentralized learning propose a convergence rate (as per Equation (3)) of:
> $$\mathcal{O}\left(\sqrt{\frac{L \Delta_0 \sigma^2}{n T}}+\sqrt[3]{\frac{L^2 \Delta_0^2\sigma^2}{(1-\beta) T^2}}+\sqrt[3]{\frac{L^2 \Delta_0^2\mathcal{H}^2}{(1-\beta)^2 T^2}}+\frac{L \Delta_0}{(1-\beta) T}\right).$$
>
>
> Here, $\beta$'s impact on the convergence rate only appears in terms formatted as $\mathcal{O}\left(\frac{1}{(1-\beta)^k}\right)$. Therefore, as long as $\frac{1}{1-\beta} \in \mathcal{O}(1)$, diminishing $\beta$ does not affect the order of the convergence rate. Hence, e.g., $\beta = \frac{1}{2}$ and $\beta = \frac{1}{100}$ would have identical convergence rates (with both yielding $\frac{1}{1-\beta} \in \mathcal{O}(1)$ and the exact value of $\frac{1}{1-\beta}$ varying by a factor of less than 2). This suggests that we cannot achieve a convergence rate better than the existing state-of-the-art solutions with $\frac{1}{1-\beta} \in \mathcal{O}(1)$. However, we observed that this limitation arises from a looseness in the conventional analysis. More specifically, the second and third terms in the above expression can be tightened such that they go to zero as $\beta$ approaches zero (see Theorem 1, and the discussion below the theorem).  This adjustment allows us to showcase the benefits of EL, which maintains a $\beta$ in $\mathcal{O}(\frac{1}{s})$.
>
> We do agree with the reviewer that parts of our analysis are similar to the standard analysis for SGD (e.g., a recursion on the function value). However, we believe the same concern applies to most of the previous works as well. After all, these methods are essentially variations of SGD with additional elements that introduce new errors, and the distinctions usually lie in the bounds on these additional error terms, as we explained above.

---

> > ### Comment · Reviewer_f2yu · 2023-08-18
> >
> > Thanks much for the response (and sorry for the delay)! I think that including some of this discussion on intuition (i.e. where the gains come from) as well as the intended settings where this learning would be more practical would be very useful. I don't have any further questions at this time.
> >
> > Thanks again!

---

### Official Review · Reviewer_VxtD · 2023-07-08

**Soundness:** 3 good
**Presentation:** 3 good
**Contribution:** 2 fair
**Rating:** 5
**Confidence:** 3

**Summary:**

This paper proposes a decentralized learning algorithm in which each node
updates its model from a set of s random nodes in a system with n > s
nodes. The authors provide a theoretical analysis of the convergence speed
and the number of transient iterations, i.e., the number of rounds required
to reach linear speedup stage. Experiments are performed for the CIFAR-10
dataset comparing the proposed two methods EL-Oracle and EL-Local with a
number of baselines that are static topologies. The comparison metrics are
accuracy, test loss, and communication volume with increasing number of
communication rounds.

**Strengths:**

A simple yet effective solution for decentralized learning. It is easily
implementable and can be easily adopted for any decentralized learning
task.

The theoretical analysis has some novel approach and is quite technical.

The paper is generally well written.

**Weaknesses:**

Title: Epidemic learning is a very confusing title. The reviewer is of the
opinion that it is an overloaded term. For example, there are many
inference problems in the context of contagions (like infectious diseases)
where the objective is to estimate information about an outbreak. Also, the
reviewer is of the opinion that this title might fail to gain the attention
of readers from the decentralized learning community. Thirdly, it is very
short and non-informative.

Theorem 1 mentions step size \gamma, which does not feature in any
expressions that appear in the statement. The expressions for \gamma in
equations (4) and (5) should be included in the statement.

Remark 1: The authors mention that they "provide convergence rate directly
for the local models". But wouldn't the expected convergence rate of the
entire global averaged model depend on the maximum of (worst case) the
convergence rates among the local models? This is not addressed.

Experiments section is weak: All the inferences are based on a single
learning problem on a single dataset. The gain in accuracy is not
significant, but again the reviewer agrees that there are benefits such as
reduction in communication rounds and volume. Yet, the gain in accuracy is
low enough to wonder what would happen if the same was used for, say a
classification task on ImageNet.

The performance for varying s: The authors only use s=7. They mention that
this is consistent with baselines. However, they could have performed a
separate experiment to study how their method performs under different
metrics for varying s. Suppose, they get very good performance for s=5
itself, wouldn't it be an interesting statement?

Disconnect between theory and experiments:

1. The authors do not make an attempt to connect theoretical results to
experiments. For example, they mention that the number of transient
iterations is an inverse function of s. There are no experiments (like the
one suggested above) to test the tightness of the theoretical bounds. A
table analogous to Table 1 for experiments would be very helpful.

2. Step size: From a theoretical perspective, the step size \gamma in
equations (4) and (5) depends on s. However, in the experiments (Appendix
D), it is obtained by only running the fully-connected topology. Some
analysis of how convergence depends on s and \gamma (as a function of
s) would be useful.

7-regular static: The authors consider only one instance of 7-regular
topology. There are many possibilities for a 96-node graph. The authors
could have considered 5-10 instances of 7-regular graphs and provided
results averaged over these topologies.

**Questions:**

There are several doubts raised in the Weaknesses section.

**Limitations:**

The authors do not bring up any limitations. No negative societal impact.

---

> ### Author Rebuttal · Authors · 2023-08-09
>
> We thank the reviewer for their comments and provide a detailed rebuttal below.
> ___
>
> > Q: Epidemic learning is not a good and informative title.
>
> Re: We can change the title to add some words about the novel element of our approach. For example, we consider changing the title to: "Epidemic Learning: Boosting Decentralized Learning with Randomized Communication".
>
> ___
> > Q: Theorem 1 mentions step size $\gamma$, which does not feature in any expressions that appear in the statement. The expressions for $\gamma$ in equations (4) and (5) should be included in the statement.
>
> Re: Thank you for the suggestion. We will move Equations (4) and (5) to the main paper, in the camera-ready version.
>
> ___
> > Q: Remark 1: The authors mention that they "provide convergence rate directly for the local models". But wouldn't the expected convergence rate of the entire global averaged model depend on the maximum of (worst case) the convergence rates among the local models? This is not addressed.
>
> Re: Note that exactly the same convergence rate as Theorem 1 holds for the global averaged model as well. In fact, proving the convergence guarantee directly on the local models requires one additional step in our analysis (see Footnote 2 on page 16 in the supplementary material). However,
> we deliberately decided to provide the convergence guarantee directly on the local models as in practice the nodes may not have access to the global averaged model. We will clarify this in the camera-ready version of the paper.
>
> ___
> > Q: All the inferences are based on a single learning problem on a single dataset. The gain in accuracy is not significant. What would happen if the same was used for, say a classification task on ImageNet.
>
> Re: We have selected the CIFAR-10 dataset for our evaluation as it is one of the most common and representative datasets in this field. Due to time and resource constraints, we are unable to run a compute-intensive dataset such as ImageNet, especially in a decentralized setting that demands significantly more resources compared to centralized learning approaches.
>
> The accuracy gain of EL on the CIFAR-10 dataset compared to the baseline is around 2\% which may or may not be considered significant.
> However, note that EL is designed to be an efficient decentralized learning algorithm, which is shown by the $1.5\times$ reduction in communication cost. Our experimental results support the theoretical foundations that EL converges faster than the baselines.
> ___
> > Q: The performance for varying s.
>
> Re:
> In response to the reviewer, we have conducted this experiment.
> We show in Figure 1 in the attached PDF of the Author Rebuttal the test accuracy for EL-Oracle and EL-Local, for $ s = 4 $, $ s = 7$ and $ s = 14 $, and when using a 7-regular static topology.
> We observe that increasing $ s $ increases the convergence speed since more models are exchanged, and results in higher test accuracy when the experiment ends.
> ___
> > Q: The number of transient iterations is an inverse function of s, but there are no experiments that test this.
>
> Re:  Less number of "transient iterations" indicates that the algorithm has a smaller second error term, which in turn causes the algorithm to perform better during the learning procedure. This is empirically confirmed in Figure 4. Please see our explanation regarding "Transient Iterations" in the global response at the top of the page for a more detailed discussion.
> ___
> > Q: Step size: From a theoretical perspective, the step size $\gamma$ in equations (4) and (5) depends on s. However, in the experiments (Appendix D), it is obtained by only running the fully-connected topology. Some analysis of how convergence depends on s and $\gamma$ (as a function of s) would be useful.
>
> Re: Our theoretical result should be interpreted as an existence result (on the step-size).
> In other words, our theory shows that there exists a step size for which the error is bounded by the expression given in the theorem, but it cannot be used to find the exact value of the step-size for the algorithm.
> Note, however, that this kind of existence result is common and constitutes a large part of the optimization literature [23].  In practice, the value of the step size should be found by hyperparameter tuning through a grid search.
> We originally performed a grid search in a fully-connected topology and used the same learning rates for other experiments, while sanity checking the performance with respect to existing works as doing a grid search for each experiment at a scale of 96 nodes is computationally intensive and we did not want to compromise on the scale.
> To address the concern of the reviewer, however, we  conducted a grid search for all the baselines in the rebuttal period, and we provide its results in the attached PDF. While we do see some minor differences in the performance for different step sizes, our main conclusion remains the same.
> Regarding the theoretical dependence of the convergence rate on $\gamma$, note that in Theorem 1, the step-size $\gamma$ is already fixed with the optimal value given in Equations (4), and (5). Also, the dependence of the rate on $s$ is explained in Section 3.2. In short, the second error term in Theorem 1 vanishes with $s$ with the rate $\mathcal{O}(\frac{1}{\sqrt[3]{s}})$.
> ___
>
> > Q: 7-regular static: The authors consider only one instance of 7-regular topology. There are many possibilities for a 96-node graph. The authors could have considered 5-10 instances of 7-regular graphs and provided results averaged over these topologies.
>
> Re: We run each experiment five times, and each run uses a different initialization seed.
> This seed also influences the generation of the 7-regular static topology.
> Therefore, we generated 5 unique instances of a 7-regular graph and presented the averaged results.
> We thank the reviewer for pointing this out and will clarify this in the experimental setup in the camera-ready version.

---

> > ### Comment · Reviewer_VxtD · 2023-08-18
> >
> > The proposed new title looks good to me.
> >
> > I still feel that the proposed method should have been evaluated on more datasets, not necessarily ImageNet. Each dataset presents its own challenge. It would be good to know if randomization consistently performs across different datasets and differet learning tasks. For example, Reference [31] use in addition to CIFAR10, a public NLP dataset as well.
> >
> > I appreciate the new set of experiments on sensitivity to s. What is the cost of increasing s? It leads to more communication, and hence, it must be slower. Is this correct?
> >
> > I am satisfied with the "7-regular static" answer.

---

> > > ### Author Response · Authors · 2023-08-19
> > >
> > > We are glad to hear that the reviewer agrees with the proposed title, and we thank the reviewer for their comments about our new experiments.
> > >
> > > **Regarding more datasets:** We agree with the reviewer that evaluating our method on different tasks and datasets would be valuable. However, we emphasize that our main contribution is theoretical, obtaining a convergence rate superior to state-of-the-art solutions. To validate our theoretical findings, we provide a proof-of-concept on the CIFAR-10 dataset, a standard dataset in the literature.
> > > However, we will conduct additional experiments for the camera-ready version to address the reviewers’ concerns. For example, the datasets included in the LEAF benchmark would be suitable for these experiments. These datasets have a natural non-IID data partitioning and are more challenging than CIFAR-10. Furthermore, the resource and compute requirements for these experiments are within our compute budget, and we see them occasionally reported in other works. We plan to conduct these experiments for the camera-ready version.
> > >
> > > **Regarding the effect of $s$:** The reviewer is correct that increasing s directly increases the communication volume. Whether this leads to an overall slowdown depends on the network of the environment where our algorithm is deployed. In data center settings where network links usually have high capacities, one can employ a high value of s. In edge settings with limited network capacities, the value of s should likely be smaller to avoid network congestion. This flexibility is, in fact, a distinguishing aspect of our approach. We will add this discussion to the camera-ready version.

---

### Official Review · Reviewer_iaf3 · 2023-07-26

**Soundness:** 3 good
**Presentation:** 3 good
**Contribution:** 2 fair
**Rating:** 5
**Confidence:** 3

**Summary:**

This paper explores decentralized learning algorithms with the aim of faster model convergence while comparable accuracy compared with conventional DL methods. The new proposed algorithm - epidemic learning (EL) - leverages a dynamically changing, randomized communication topology to train a machine learning model in DL environment. The paper provides theoretical analysis which shows that the EL algorithm surpasses the best-known static and randomized topologies in terms of convergence speed, w.r.t. two key properties: linear speed-up and transient iterations. The experimental results show that the proposed EL-oracle and EL-local achieve quicker convergence than baselines.

**Strengths:**

This paper is studying a very interesting and important problem - DL algorithms. The paper is well written, which is easy to follow. The proposed EL algorithm is technically sound. The theoretical analysis and experimental results show its effectiveness compared to baselines.


**Weaknesses:**

A major concern is that the proposed EL algorithm is not significantly different with semi-dynamic and time-varying and randomized topologies (they are introduced in related work section). Especially, gossip learning (GL) is like a special case of EL-local with s = 1.

A minor comment: Figure 3 should be improved by giving label for x-axis. Also there is no full words for the first abbr "CDF".


**Questions:**

1. Compared with semi-dynamic topologies, and time-varying and randomized topologies, what's the signaficance of EL? Have any of the previous works provided theoretical analysis on convergence rate?
2. What does "the convergence of GL on non-convex problems remains uncertain" mean?
3. Is GL a special case of EL-local with s=1? In the experiment, have you done sensitivity analysis on s values?

**Limitations:**

Not applicable.

---

> ### Author Rebuttal · Authors · 2023-08-09
>
> We thank the reviewer for their comments and provide a detailed rebuttal below.
> ___
>
> > Q: A major concern is that the proposed EL algorithm is not significantly different with semi-dynamic and time-varying and randomized topologies (they are introduced in the related work section). Especially, gossip learning (GL).
>
> Re: We agree with the reviewer that the idea of running SGD over time-varying and randomized topologies existed in some previous works, including GL (as we outlined in the related work section).
> However, our work is the first, to our knowledge, that provides a theoretical foundation highlighting the advantages of randomized communication for decentralized learning.
> In fact, our main contribution is to prove that an adaptation of SGD over these randomized networks can outperform the state-of-the-art decentralized learning schemes.
> Hence, we believe the simplicity and effectiveness of our algorithm only enhance its practical relevance for the community.
> ___
> > Q: What does "the convergence of GL on non-convex problems remains uncertain" mean?
>
> Re: With this sentence, we mean that the convergence of GL on non-convex loss functions has not been theoretically proven yet in the works that introduce and evaluate GL [15-16]. In fact, these works are mainly empirical and do not provide any convergence guarantee.
> ___
> > Q: Is GL a special case of EL-local with s=1?
>
> Re: While at a high level, EL-Local with $s=1$ may look very similar to GL, there are some subtle differences, mainly in the way the aggregation proceeds.
> GL is inspired by the gossip protocols [47], and involves two parallel steps.
> At each time step, a node sends a message (a model in GL) to one randomly chosen node ($s=1$).
> Independently, on receiving a single message, a node aggregates the received model from this message with its own model and updates the aggregated model.
>
> In contrast, EL-Local is designed to be aligned with existing decentralized learning algorithms (D-PSGD), with some basic modifications.
> EL-Local proceeds in rounds where first a node updates its local model on its local dataset and sends its model to $s$ randomly-chosen nodes.
> Next, the node waits to receive the models from other nodes in the current round.
> On receiving one or more models in the ongoing round, the node aggregates them all together.
> Even if each node chooses exactly one random node to send the message to ($s=1$), it can receive messages from multiple nodes due to randomness.
> In other words, with $s=1$, if a node receives multiple models in a round, nodes in GL will aggregate the received model with its local model for each received model separately, whereas in EL-Local, there will be a single model update per round, and all the received models from that round are aggregated together.
>
> From a theoretical point of view, this subtle difference plays an important role as it ensures the local models of the nodes stay close to each other during the training (Lemma 2), which is essential for the proof of convergence.
>
> In summary, while GL shares some high-level similarities with our approach, GL is not a special case of EL-Local under $ s=1 $.
> ___
> > Q: In the experiment, have you done sensitivity analysis on s values?
>
> Re: We have not done a sensitivity analysis on $ s $ for the original submission, but in response to the reviewer we have conducted this experiment in the rebuttal period.
> We show in Figure 1 in the attached PDF in the Author Rebuttal the test accuracy for EL-Oracle and EL-Local, for $ s = 4 $, $ s = 7$, and $ s = 14 $, and when using a 7-regular static topology.
> We observe that increasing $ s $ increases the convergence speed since more models are exchanged, and results in higher test accuracy when the experiment ends.

---

> > ### Comment · Reviewer_iaf3 · 2023-08-16
> > **I have read the response**
> >
> > Thanks for the response to my question. Most of my questions have been answer except Question 3. What's the result if you compare EL-Local with s=1 and GL method?

---

> > > ### Author Response · Authors · 2023-08-17
> > >
> > > We are glad to hear that most of the reviewer’s questions have been addressed.
> > >
> > > As we answered Question 3 in the rebuttal, could the reviewer specify in which aspect they look for further explanation?  We have explained why GL is not a special case of EL-Local with s=1 (first part of the question), and we have also conducted a sensitivity analysis on the values of s, which can be found in the attached PDF document (second part of the question).
> > >
> > > The reviewer might be asking for additional experimental results, comparing GL's achieved test accuracy and communication efficiency with EL and other baselines. Despite the algorithmic similarities between EL-Local and GL, we have not included GL as an experimental baseline because GL operates asynchronously, whereas EL-Local is a synchronous algorithm. This difference complicates a fair and clean comparison. Specifically, the performance of GL highly depends on the underlying node and network characteristics, and it can be made arbitrarily bad if network links are slow and heterogeneous. Under these circumstances, the local models might drift away from each other. Therefore we did not include this result in Figure 4. _If the reviewer believes this result is valuable, we can consider adding this to the camera-ready version._
> > >
> > > A similar argument exists on the theoretical side. There is no theoretical convergence rate for GL in prior work, and analyzing this algorithm requires additional assumption that prevents the local models from drifting away from each other. We consider this analysis beyond the scope of our paper.

---

> > > > ### Comment · Reviewer_iaf3 · 2023-08-18
> > > > **No more questions**
> > > >
> > > > My questions have been resolved.

---

> > > > > ### Author Response · Authors · 2023-08-18
> > > > >
> > > > > We thank the reviewer for their reply and we will add a discussion to the camera-ready version. As the concerns raised by the reviewer have been addressed, and the initial rating was only 5 (i.e., Borderline Accept), we sincerely hope that the reviewer considers adjusting their rating.

---

### Author Rebuttal · Authors · 2023-08-09

Firstly, we would like to thank all reviewers for the thorough and insightful comments on our submission. We appreciate the detailed feedback and the points raised, which have offered some valuable new insights.

Below we address two comments that were raised by multiple reviewers.

## Transient Iterations

As multiple reviewers raised concerns regarding the concept of "transient iterations" and its connection to our experimental results, we provide here a clarification:

To clarify, in the context of decentralized learning schemes, the convergence analysis often involves three error terms (as in Theorem 1). Generally, the first error term $\mathcal{O}\left(\frac{1}{\sqrt{nT}}\right)$ is attributed to the stochastic noise in gradient computation during the SGD step. The last error term $\mathcal{O}\left(\frac{1}{T}\right)$ is caused by the initial error in the function value. Notably, these two error terms are generally identical for different schemes since they are not directly related to decentralization. In fact, they even exist when running a centralized SGD on a single node.

In contrast, The second error term $\mathcal{O}\left(\frac{1}{T^{2/3}}\right)$ is the additional error introduced by decentralization and arises due to model drift: different nodes computing gradients on different local models at each step. This term is directly influenced by the efficiency of the mixing within the communication network. Hence, this term is the primary term to consider while comparing different decentralized learning schemes. The first term, despite having a stronger dependence on $T$, does not influence the comparison, as it is always fixed regardless of the communication network, be it say a sparse ring or a complete graph (see Equation (3) in the paper). Also, usually, the number of iterations required for the first term to become dominant is very large and it could even be more than the total number of learning iterations in a practical DL setting.

To capture this phenomenon and facilitate a more straightforward comparison of the second error term, the concept of "transient iterations" has been proposed in previous works. This concept does not merely imply that an algorithm is faster only during the initial stages of learning. Instead, it indicates that the algorithm has a smaller second error term, which in turn causes the algorithm to perform better during the learning procedure.

Considering this perspective, Figure 4 does not contradict the theoretical statement about the superiority of convergence in transient iterations. Instead, it suggests that EL should have a more favorable second error term, resulting in its out-performance throughout the learning process that matches our theoretical findings.

## Additional Experiments

As requested by multiple reviewers, we have added two additional experiments which results can be found in the attached PDF below. Our first additional experiment explores the effect of different values of $ s $, i.e., the number of neighbors we send a model to. Our second additional experiment involves tuning of the step-size $\gamma$. We hope this addresses the concerns of the reviewers.

---

> ### Author Response · Authors · 2023-08-16
>
> Dear reviewers and AC,
>
> We appreciate your time and effort in reviewing our paper and we hope that your concerns have been addressed by our rebuttal. As the discussion period ends soon, we hope that you would consider participating in an interactive discussion to resolve any further concerns regarding our paper.
>
> If our responses have clarified your concerns, we kindly ask you to re-evaluate your recommendation.
>
> With kind regards,
> The authors

---

### Decision · Program_Chairs · 2023-09-21

**Decision:**

Accept (poster)

**Comment:**

The paper provides an improved convergence guarantee for distributed optimization that improves on previous work in terms of network scaling. The contribution of the work is clear and all reviewers believe the paper should be accepted.